# Deep learning driven biosynthetic pathways navigation for natural products with BioNavi-NP

Shuangjia Zheng[1,2,3,6,7], Tao Zeng [1,7], Chengtao Li[3], Binghong Chen[4], Connor W. Coley [5], Yuedong Yang [2✉] & Ruibo Wu [1✉]

The complete biosynthetic pathways are unknown for most natural products (NPs), it is thus valuable to make computer-aided bio-retrosynthesis predictions. Here, a navigable and user-friendly toolkit, BioNavi-NP, is developed to predict the biosynthetic pathways for both NPs and NP-like compounds. First, a single-step bio-retrosynthesis prediction model is trained using both general organic and biosynthetic reactions through end-to-end transformer neural networks. Based on this model, plausible biosynthetic pathways can be efficiently sampled through an AND-OR tree-based planning algorithm from iterative multi-step bio-retrosynthetic routes. Extensive evaluations reveal that BioNavi-NP can identify biosynthetic pathways for 90.2% of 368 test compounds and recover the reported building blocks as in the test set for 72.8%, 1.7 times more accurate than existing conventional rule-based approaches. The model is further shown to identify biologically plausible pathways for complex NPs collected from the recent literature. The toolkit as well as the curated datasets and learned models are freely available to facilitate the elucidation and reconstruction of the biosynthetic pathways for NPs.

[1] School of Pharmaceutical Sciences, Sun Yat-sen University, Guangzhou 510006, China. [2] School of Computer Science and Engineering, Sun Yat-sen University, Guangzhou 510006, China. [3] Galixir, Beijing, China. [4] College of Computing, Georgia Institute of Technology, Atlanta, GA, USA. [5] Department of Chemical Engineering, Massachusetts Institute of Technology, Cambridge, MA, USA. [6] Present address: School of Computer Science and Engineering, Sun Yat-sen University, Guangzhou 510006, China. [7] These authors contributed equally: Shuangjia Zheng, Tao Zeng. ✉email: yangyd25@sysu.edu.cn; wurb3@sysu.edu.cn

To date, more than 300,000 natural products (NPs) have been discovered and catalogued in libraries such as Dictionary of Natural Products[1] (DNP) and Super Natural II[2]. Remarkably, this vast chemical space of NPs is reachable from a few dozen, simple building blocks[3,4]. According to the classes of those building blocks, there are correspondingly four well-known biosynthetic pathways for major classes of NPs and their hybrids, including: (1) the AA/MA (acetic acid and malonic acid) pathway that produces fatty acids, phenols, and polyketides; (2) the MVA/MEP (mevalonic acid or methylerythritol phosphate) pathway that generates terpenoids and steroids; (3) the CA/SA (cinnamic acid or shikimic acid) pathway that yields flavonoids, phenylpropanoids, lignans, and coumarins; and (4) the AAs (amino acids) pathway that constructs alkaloids and peptides including ribosomally synthesized and posttranslational modified peptides

(RiPPs) and non-ribosomal peptide (NRPs). Unfortunately, as shown in Fig. 1a, only about 33,000 enzymatic reactions have been characterized and confirmed, corresponding to fewer than 30,000 NPs serving as a substrate or product (data was collected from public sources such as MetaCyc[5], KEGG[6] and MetaNetX[7]). That is, complete biosynthetic pathways including all intermediates are not established for most of the hundreds of thousands of known NPs. Accordingly, there is a strong desire to reveal the biosynthetic pathways from essential building blocks to target NPs (namely the native NPs biogenesis).

NPs have been noted to exhibit larger structural diversity than fully synthetic molecules and exist in a distinct chemical space[8]. As a result, NPs play a significant role in drug discovery: more than 60% of FDA-proved small molecule drugs are NPs or their derivatives[9]. NPs are often the best option for seeking novel

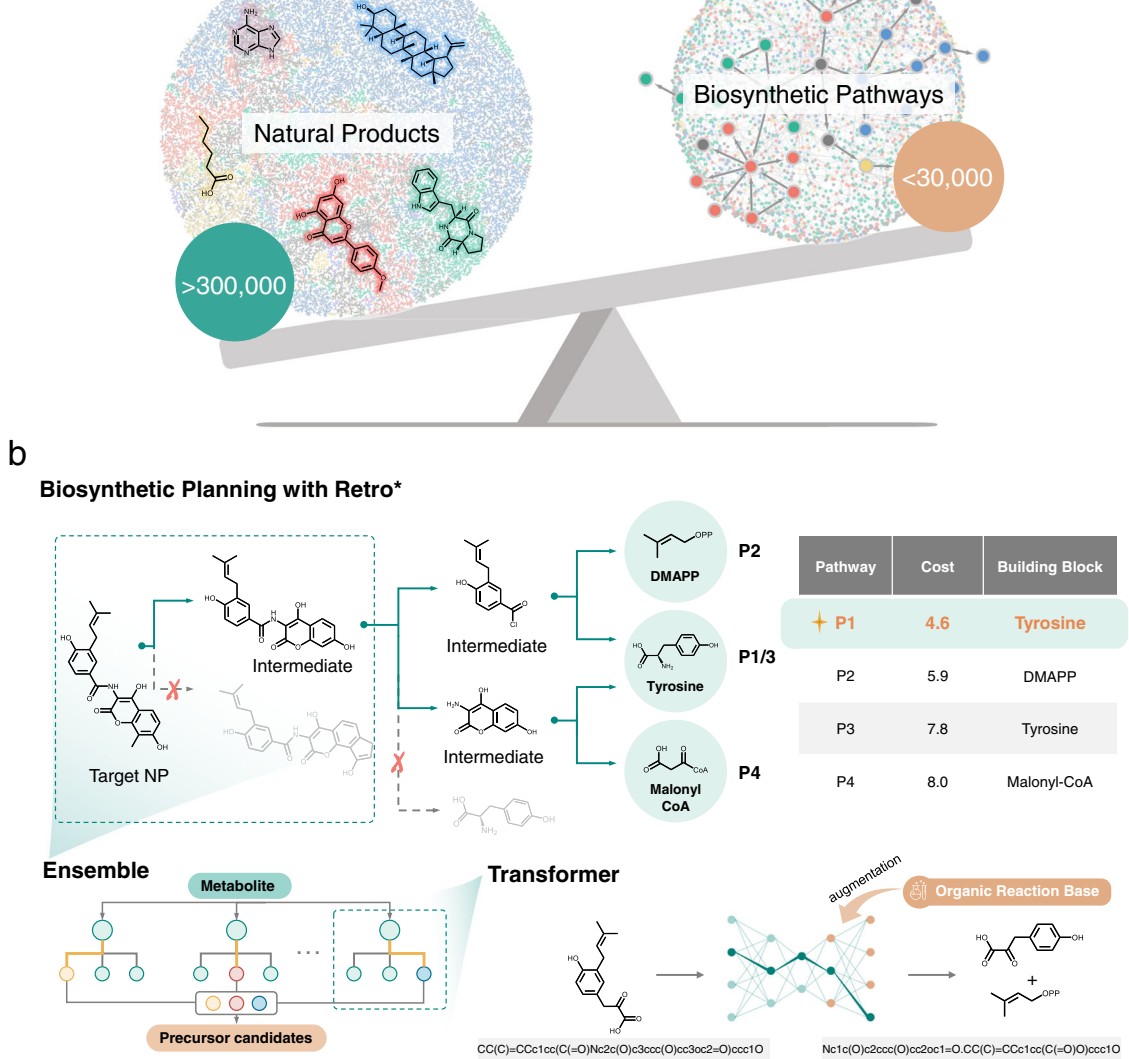

**Fig. 1 The motivation and overview of BioNavi-NP. a** The vast natural products and rare biosynthetic pathways reported to date. Natural products were collected from DNP[1] and visualized by TMAP[73] (left). Biosynthetic reactions were collected from MetaCyc[5], KEGG[6] and MetaNetX[7], and the network was visualized by Cytoscape[74] (right). The structures were represented by the nodes and similar structures converged. The edges and arrows in the biosynthetic network represent the structural transformation. Fatty acids and others from the AA/MA pathway were colored yellow. Terpenoids and steroids from the MVA/MEP pathway were colored blue. Flavonoids and others from the CA/SA were colored red. Alkaloids and others from the AAs pathway were colored green. Others, such as nucleic acids and some hybrid-origin compounds, were colored black. **b** The protocol of BioNavi-NP to explore biosynthetic pathways of target natural product. We trained the transformer neural networks by combining biosynthetic and organic reactions, and four models trained with different hyperparameters form the ensemble model, which was finally used to make the single-step prediction (see details in Methods, Supplementary Figs. 1 and 2).

bioactive templates in human health, as many of them are critical factors for regulating intrinsic biofunctions, especially in plants[10,11]. However, obtaining commercially-relevant quantities of NPs can be a major obstacle to their therapeutic translation, since NPs are usually expressed in low abundances in natural sources, and thus conventional extraction approaches are inefficient and environmentally unfriendly in most cases. Meanwhile, many highly valuable NPs have complicated structures, making total synthesis difficult and time-consuming. As famously exemplified by the heterologous biosynthesis of the arteannuinic acid[12], biosynthesis and semi-synthesis of complex natural products have become more popular and influential strategies in the past decade because of the theoretical advantages of lower cost and higher yield. Nevertheless, it remains challenging to reconstruct a heterologous biosynthetic pathway from its native pathway. Therefore, computer-aided tools are needed for the retro-biosynthesis analysis of target NPs, especially for the exploration of non-native biosynthetic pathways.

To date, many efforts have been made in the field of retro-biosynthesis[13–15], which can be roughly divided into two categories of methods: knowledge-based and rule-based approaches (Supplementary Table 1 provides an overview on popular methods). The knowledge-based approaches enumerate possible biosynthesis routes according to the existing reaction databases (such as MetaCyc[5] and KEGG[6]) and rank the suggested routes through scoring functions such as chemical similarity[16,17] and chassis[18]. For complex NPs, these methods are often not applicable when the reactions of their biosynthetic pathways are not included in those databases. The template- or rule-based models (e.g., RetroPath 2.0[19], RetroPathRL[20], and so on[21,22]) match the query molecule to a collection of generalized reaction rules, i.e., subgraph patterns of molecules that highlight changes during biochemical reactions. The rules are either summarized manually by experts[23,24] or extracted automatically from the reaction databases[25]. Although rule-based methods have led to promising results in retro-biosynthesis, a few challenges remain. First, their ways to formulate expert-approved rules are complicated and time-consuming[25]. Second, the degree of generality/specificity of curated rules may lead to invalid or incomplete proposals[26]. Third, they are fundamentally unable to predict reactions beyond the rule databases[27].

Recently, the development of deep learning methods has made it possible to predict reactions without rules, where molecules could be represented as strings (e.g., SMILES[28]) as input into sequential models such as recurrent neural networks[29] and transformer neural networks[30]. Such techniques have been utilized to predict the products of both organic[31,32], and enzymatic reactions[33,34] by giving reacting substrates as input, or the reverse for retrosynthesis prediction task[35,36]. These rule-free models have shown better performance and greater generalization potential than rule-based models in many cases[35]. Based on the single-step retrosynthesis prediction, the retrosynthetic pathways can be planned through searching techniques (e.g., Monte Carlo tree search, MCTS), but MCTS-based methods[20,37,38] are of limited efficiency due to the required expensive online reward estimation. More recently, Chen et al.[39] reported a deep learning-guided AND-OR tree-based searching algorithm called Retro*, and demonstrated improved planning efficiency and solution quality. Nevertheless, multi-step planning algorithms have not yet been applied to NPs retro-biosynthetic planning, mainly due to its much less available data, greater number of steps, and higher branching ratios in the biosynthetic pathways.

Herein, we present BioNavi-NP as a practical tool to propose NP biosynthetic pathways from simple building blocks in an optimal fashion. As depicted in Fig. 1b, we first train transformer neural networks to generate the candidate precursors for a target NP. Through data augmentation[40] and ensemble learning[41], our best model achieves a top-10 prediction accuracy of 60.6% on the single-step biosynthetic test set, 1.7 times more accurate than the previous rule-based model. Based on the single-step model, we further develop an automatic retro-biosynthesis route planning system (BioNavi-NP) through the deep learning-guided AND-OR tree-based searching algorithm, which can solve the combinatorial number of options caused by the branches of the synthetic pathway. As a result, BioNavi-NP successfully identify biosynthetic pathways for 90.2% of 368 test compounds and recover the reported building blocks as in the test set for 72.8%, demonstrating its potential for bio-retrosynthetic pathway elucidation or reconstruction. For each biosynthetic step in the multi-step bio-retrosynthetic routes, we further evaluate the plausible enzymes through enzyme prediction tools, Selenzyme[42] and E-zyme 2[43]. All outputs of the BioNavi-NP can be visualized by an interactive website (http://biopathnavi.qmclab.com/), where the predicted reaction pathways are sorted by the computational cost, length, and organism-specific enzyme.

## Results

**Single-step evaluation**. Multi-step retro-biosynthesis planning is based on the backward search performed through iterative single-step retrosynthesis predictions. Therefore, it is critical to achieve a reliable prediction of single-step precursors at each step. Our first task is to compare various architectures and training modes to determine an optimal model. To train our model, we first curated a biosynthesis data set from the public database called BioChem (see Methods), containing 33710 unique pairs of precursors and metabolites. From the dataset, we randomly selected 1000 pairs as the test set, 1000 pairs as the validation set, and the remaining as the training set. Following the previous retrosynthesis works[31,35,44], we evaluated the performance on the test set using top-n accuracies, defined as the percentages of correct instances among top-n predicted precursors. Considering the complexity of biosynthesis, we further expanded the training set by retrieving 62370 organic reactions similar to these biochemical reactions from USPTO[45], the largest organic chemical reaction library available. This strategy was inspired by transfer learning[40], where a sufficient amount of relevant data helps to improve model robustness by learning general patterns and avoiding over-fitting. The larger data set of 60 K natural product-like reactions is named USPTO_NPL (see Methods for more details).

As summarized in Table 1, the transformer model directly trained on the BioChem training set with 31,710 reactions achieved top-1 and 10 accuracies of 10.6% and 27.8%, respectively. The correct handling of stereochemical information contributed to the prediction of biosynthetic reaction, as the removal of chirality from the reaction SMILES decreased the top-10 accuracy from 27.8% (BioChem) to 16.3% (BioChem (w/o chirality)). When 60 K organic reactions involving natural

**Table 1 Performance of single-step models by different training strategies.**

| Training strategy | top-N accuracy (%), $N =$ | | | |
|---|---|---|---|---|
| | 1 | 3 | 5 | 10 |
| USPTO_NPL | 0 | 0 | 0 | 0 |
| BioChem (w/o chirality) | 7.6 | 11.1 | 13.9 | 16.3 |
| BioChem | 10.6 | 20.1 | 24.5 | 27.8 |
| BioChem + USPTO_NPL | 17.2 | 30.2 | 41.9 | 48.2 |
| BioChem + USPTO_NPL (ensemble) | 21.7 | 42.1 | 52.4 | 60.6 |
| BioChem + USPTO_NPL (seq2seq) | 10.9 | 21.3 | 30.8 | 37.1 |
| RetroPathRL | 20.6 | 30.5 | 36.8 | 42.1 |

product-like compounds were joined for training (BioChem + USPTO_NPL), the top-1 and 10 accuracies of the model raised to 17.2% and 48.2%, respectively. The large increase in accuracies by data augmentation indicated that organic reaction expertise was helpful for accurately predicting biological steps. Meanwhile, the model trained solely on NP-like organic reactions (USPTO_NPL) did not make any correct predictions of biosynthetic precursors, indicating that biosynthetic NPs and chemically synthesized compounds share commonalities, but represent two distinct structural spaces and distinct sets of reaction types. Moreover, these results explained why existing organic retrosynthesis tools cannot be directly used for biosynthesis prediction. An additional improvement in accuracy was obtained by ensembling four optimal transformer models with different training steps (BioChem + USPTO_NPL(ensemble)), leading to top-1 and 10 accuracies of 21.7% and 60.6%, respectively. This is expected since the ensemble procedure[41] can reduce the variance of different models' random initializations and improve robustness.

Table 1 also showed that our model consistently outperformed the state-of-the-art rule-based biosynthesis model, RetropathRL[20], with an increase by up to 1.1% and 18.5% in terms of top-1 and 10 accuracies, respectively. This result demonstrated the power of the deep learning-based methods for the prediction of biosynthetic processes, while the higher accuracy of BioChem + USPTO_NPL than BioChem + USPTO_NPL (ses2seq) demonstrated the value of the attention mechanism by transformer. Hereafter, we would use the BioChem + USPTO_NPL (ensemble) as the basic single-step prediction model if not specially mentioned.

**Internal testing for multi-step planning**. Next, we investigated the performance of different multi-step planning strategies. We integrated the above trained single-step prediction ensemble model to navigate the multi-step bio-retrosynthesis planning and named it BioNavi-NP. To evaluate the searching performance, we randomly selected 368 diverse NPs with complete biosynthetic pathways from the BioChem dataset, and compared different methods in terms of their abilities to: (i) predict biosynthesis routes that terminate in allowable building blocks (success rate or solution rate), (ii) to correctly find the reported pathway exactly as it appears in the knowledge base (hit rate of pathways), and (iii) to correctly recover the building blocks used in the known biosynthetic pathway (hit rate of building blocks). We introduced the hit rate of building blocks because the pathway of ground truth is not necessarily unique. There may be multiple pathways between a natural product and its building blocks. Even if a candidate pathway is not the ground truth, the candidate is also meaningful as a non-native biosynthetic pathway or an inspiration for pathway reconstruction and design. To ensure a fair comparison, all deep learning models were limited to 100 iterations and 10 expansions, while RetroPathRL[20] was set to 1000 iterations and 10 expansions following its default settings. Note that the number of expansion represents the top-N metabolites that the model will predict in every single step, i.e., top-N in Table 1. The maximum depth of the predicted pathways was set to 10. By default, 40 building blocks (called the core library, Supplementary Fig. 3) were used, and the pathway search stopped once it met any of the building blocks.

As summarized in Table 2, BioNavi-NP outputted potential biosynthetic pathways for 332 out of 368 target NPs (90.2% success rate), a large improvement over the state-of-the-art retro-biosynthesis pathway prediction tool RetroPathRL[20] (52.7%). Meanwhile, BioNavi-NP achieved 56.0% and 24.7% for the hit rates of building blocks and pathways, remarkably outperforming RetroPathRL (4.8% and 3.8%), respectively. It should be noted that RetroPathRL contains the reaction rules extracted from

MetaNetX[7], while all the internal test NPs were included in MetaNetX. When changing the searching algorithm to MCTS, BioNavi-NP (MCTS) achieved a substantially lower success rate of 34.8%, indicating the importance of searching strategy to manage the highly-branched multi-step search. As users may prefer well-known or user-specific building blocks in practice, we set the building blocks of ground truth as the user-defined building blocks (UDBs), and terminated predictions only with the UDBs. As expected, the BioNavi-NP_UDB model increased both the hit rates of building blocks and pathways (72.8% and 26.1%, respectively) while decreased the success rate (74.7%). The same trend was observed between RetroPathRL and RetroPathRL_UDB.

To guide the reconstruction of the biosynthetic pathway, it is important to explore more than one pathway for each target NPs. Therefore, in addition to the abovementioned hit rate of pathways, we also counted the average number of predicted pathways as a direct indicator for the tool's practical use. When selecting the output option as top-5 (see Table 2), BioNavi-NP predicted an average of 4.9 pathways. The exploration ability was further confirmed by the longest length (six) of the pathways predicted by BioNavi-NP in comparison to the three by RetroPathRL and BioNavi-NP (MCTS), indicating the ability of our model to produce more hypothetical pathways with more complexity.

In addition, we compared the exploration abilities and hit rates over different NP families. For the five categories of natural products in the test set (including AA/MA, AAs, CA/SA, MVA/ MEP and Others, as shown in Fig. 2a), BioNavi-NP achieved the highest hit rates of pathways (54.1%) and building blocks (94.6%) for the AA-MA category (Fig. 2b), and following were CA/SA and MVA/MEP category. The AAs category had the lowest hit rates of 18.3% and 46.2%, respectively. This is likely attributed to the diverse building blocks of complex structures, especially the RiPPs and NRPs, which often consisted of more than three building blocks (amino acids or even non-proteinogenic amino acids). It is difficult for BioNavi-NP to decompose these complex structures into several fragments due to the lack of such data for training. The model performance on NPs from different kingdoms was further investigated (Supplementary Fig. 5), where the NPs in the internal set were manually divided into plant, bacteria, fungi, animal and others (such as some algae like Bacillariaceae). The results showed that the model achieved the highest hit rate of pathway and building blocks (30.7% and 82.0%, respectively) for the plant kingdom. This is consistent with our results at the pathway level because the plant NPs mainly fall into the MVA/MEP and CA/SA categories (Supplementary Table 2). By contrast, NPs from the fungi and others had lower hit rate of building blocks (46.1% and 41.7%, respectively), mainly due to the fact that the majority of them in the internal set are NRPs derived from the AAs pathway.

**External testing for multi-step planning**. To further evaluate the generalizability of BioNavi-NP, we collected 25 unseen NPs compounds (external cases) from recent publications that did not appear in the training set or internal test set. Out of these, 22 cases successfully found at least one candidate pathway with a success rate of 88%, roughly the same as the 90.2% in the internal test. For the 3 remaining cases, incomplete results were still outputted for analysis. 17 cases had correctly identified at least one building block with the hit rate of building blocks of 68%, higher than the 56% in the internal test (five representative cases were shown in Supplementary Fig. 6). The complete pathways of all these cases can be found in Supplementary Figs. 7–31, and a web version is provided at http://biopathnavi.qmclab.com/case.html. These suggest that BioNavi-NP can be an effective tool for

**Table 2 Comparison of performance among different models for the test set.**

| Methods | Success rate | Hit rate of building blocks | Hit rate of pathways | Longest length | Avg. solution[a] | Time (h)[b] |
|---|---|---|---|---|---|---|
| BioNavi-NP (MCTS) | 34.8% | 16.3% | 1.9% | 3 | 1.0 | 92 |
| RetroPathRL | 52.7% | 4.8% | 3.8% | 3 | 2.8 | 2 |
| BioNavi-NP | 90.2% | 56.0% | 24.7% | 6 | 4.9 | 18 |
| RetroPathRL_UDB | 10.8% | 5.1% | 4.1% | 3 | 2.8 | 3 |
| BioNavi-NP_UDB | 74.7% | 72.8% | 26.1% | 6 | 4.9 | 28 |

UDB user-defined building blocks.
[a]Denotes the average number of pathways found, only the top-1 result is supported by the MCTS algorithm, while for RetroPathRL, it outputs all pathways it can find. The output option for Retro* is set as top-5 (default is top-10).
[b]It is an about 4-times computational time for outputting top-10 in comparison to top-5, that is, the time consuming of BioNavi-NP (if only requesting the top-3) is comparable to RetroPathRL (the average number of pathways returned by RetroPathRL is close to 3).

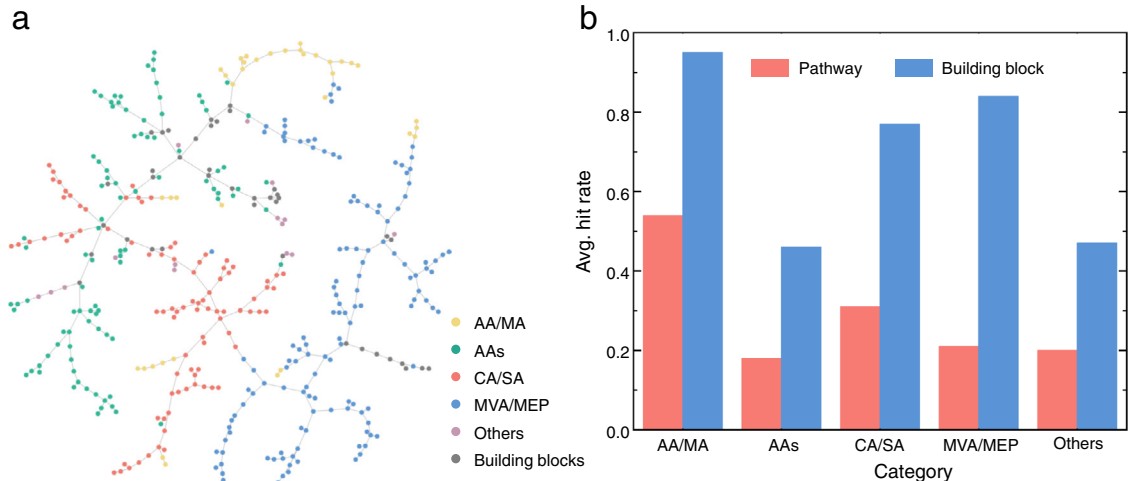

**Fig. 2 The distribution and performance of internal test set. a** The chemical space of internal cases in each NPs category and building blocks. The clustering and visualization of chemical space were realized by TMAP[73] using structural molecular fingerprints[75]. The nodes represent the structures and similar structures converge and are clustered on the same branch. The comparison of chemical space for the training set, internal cases and external cases is provided in Supplementary Fig. 4. **b** The BioNavi-NP's performance within each NP category. Source data are provided as a Source Data file.

retro-biosynthesis analysis in terms of its abilities to find building blocks and to enumerate hypothetical biosynthetic pathways of target NPs.

For a direct comparison with existing works, we tested BioNavi-NP on two benchmark datasets (namely LASER and Golden dataset) used by RetroPathRL[20], which were used to evaluate the ability to predict biosynthesis routes (success rate) and the pathways (hit rate of pathways). For a fair comparison, we used the same building block library (extended library) containing 437 available precursor metabolites extracted from iML1515 model[46], as used in RetroPathRL. The parameters of BioNavi-NP were set to a maximum of 100 iterations and 10 expansions. As shown in Supplementary Table 3, our model achieved a hit rate of 94.7% (144/152) on the LASER test set, outperforming RetroPathRL (83.6%) and RetroPath 2.0 (77.6%). In terms of the hit rate of pathways, BioNavi-NP predicted exactly the ground-truth pathway for 13 cases for a total of 20 cases in the Golden dataset, comparable to 15 cases by RetroPathRL. Note that to achieve these results, RetroPathRL needed to increase the iteration numbers from 1000 to 10,000 ~ 15,000, and thus takes 9 times longer than BioNavi-NP (123 h for RetroPathRL vs. 13.5 h for BioNavi-NP). Hence, our method presents both high accuracy and high efficiency. Besides, since most of the metabolites in the extended library are well-known precursors and lie in the cytosol compartment of microbial strains, we also included these 437 metabolites as an option in our web server.

**Web deployment and case study.** BioNavi-NP was implemented in a webserver for the convenient redesign of biosynthetic pathways. As shown in Fig. 3a, like a widely-used organic synthesis planning tool ASKCOS[47] did, only the structure of the target molecule is strictly required, but the default settings and list of building block can be modified as needed. The total number of output pathways is influenced by several options, especially the "pathway_top_k" (default is top-10). The length of the pathways is affected by the "Max_depth" (default is 10). One can increase the number to a maximum of 20 upon request. These default settings (Fig. 3a, see more details in Supplementary Method 1) are recommended to balance the computational time and accuracy. Predicted biosynthetic pathways are displayed in an interactive network, in which the target molecule can be traced back to the potential building blocks through several pathways along with the predicted cost of each step. Known reactions and intermediates are matched with MetaNetX[7] and PubChem[48], respectively, and highlighted in the interactive network. The above annotations as well as the number of reactions and intermediates of each pathway are also summarized in a table, where users can re-rank the pathways as needed. The typic result panel is shown in Supplementary Fig. 32. For a predicted biosynthetic pathway, it is crucial to identify the enzymes that will enable the reaction to take place. Several tools exist to select candidate enzymes for novel reactions, including Selenzyme[42], E-zyme 2[43], BridgIT[49], and BRENDA[50]. We introduced Selenzyme to search for candidate enzymes for

a

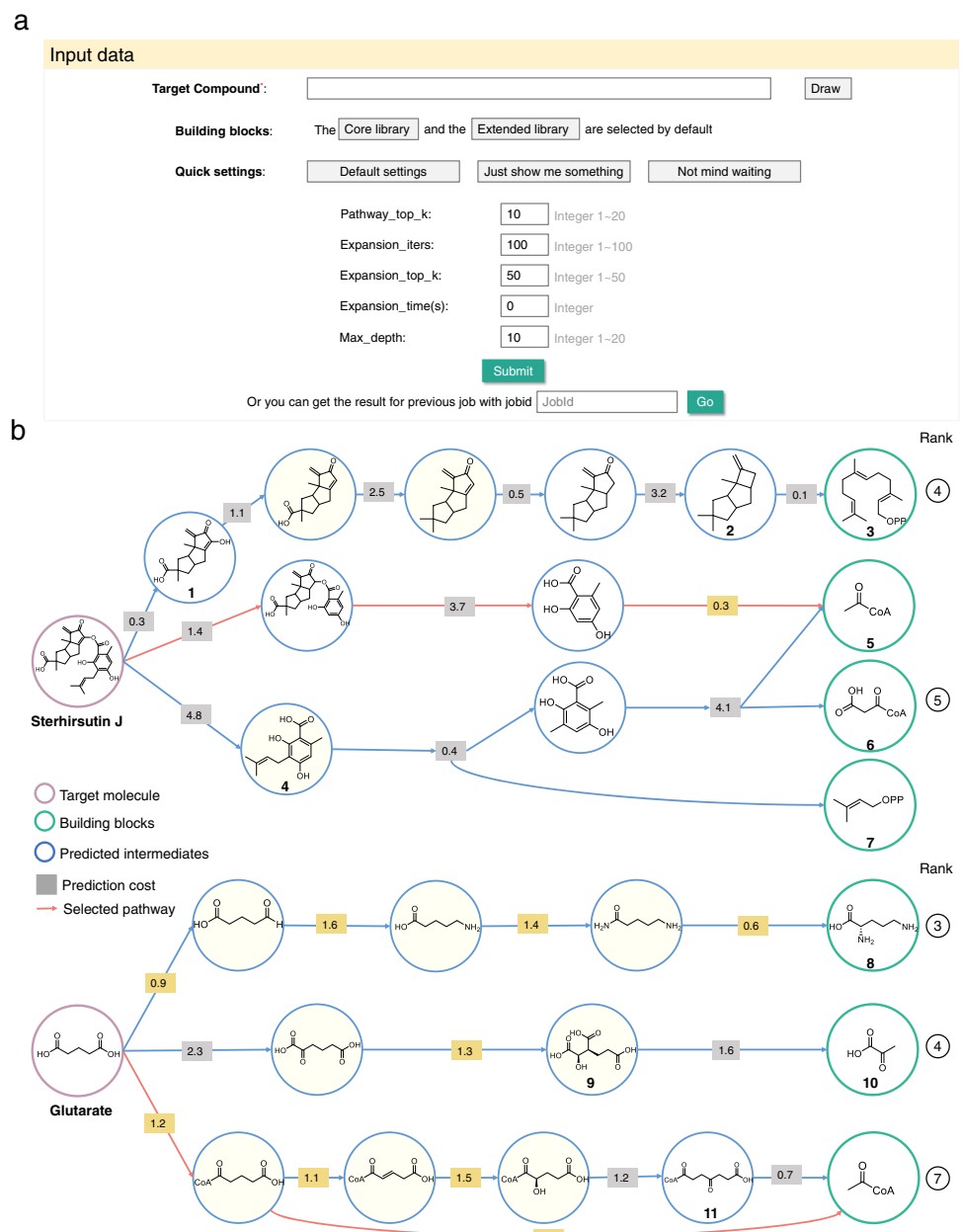

**Fig. 3 The interface and output of BioNavi-NP webserver. a** The input interface and of BioNavi-NP webserver. **b** The selected pathways of two examples (sterhirsutin J and glutarate) predicted by BioNavi-NP. Herein the outputs are redrawn to be clear (the raw output provided in Supplementary Figs. 27 and 33), and some candidate pathways with the order of ranks attached at the end, and the top 1 pathway is colored in red. The known intermediates and reactions are highlighted in yellow. There is also an option to select which pathway to be highlighted or shown. The cost of each reaction step is reflected by the confidence score (smaller prediction cost means higher reaction probability, see Methods), and the total cost of the pathway is used to rank the network by default.

specific reactions because of its lightweight RESTful service and additional biological metadata. As a result, pathways output by BioNavi-NP can be further ranked by reaction similarity and the taxonomic distance between the organism of enzyme and user-defined organism. In addition, the hyperlink and required input chemical information of E-zyme 2 are also provided for enzyme prediction as an alternative. To illustrate the use of the webserver, we selected the sterhirsutin J and glutarate for the case studies. The server returned 6 and 10 candidate pathways, respectively. Figure 3b shows a subset.

Sterhirsutin J is a sesquiterpenoid derivative firstly isolated from the culture of *Stereum hirsutum* that has shown cytotoxicity against K562 and HCT116 cell lines[51]. As shown in Fig. 3b, sterhirsutin J

were decomposed into a hirsutane-type sesquiterpene (intermediate **1**) and colletorin D acid (intermediate **4**) to lead the fourth and fifth candidate pathways, respectively, where the fourth candidate pathway was ultimately traced back to the building block farnesyl diphosphate (**3**), while the fifth candidate way was a hybrid biosynthesis style originated from the acyl CoA (**5**), malonyl CoA (**6**), and dimethylallyl diphosphate (**7**). Both the fourth and fifth candidates are confirmed biosynthetic pathways according to the previous studies[52,53]. This case shows us that BioNavi-NP is capable of dealing with complex structures including those derived from the hybrid pathway and to trace them back to essential building blocks.

Glutarate (also called 1,5-pentanedioic acid) is an important raw material for the chemical industry, but biobased production

of glutarate suffers from low titers[54]. Herein using BioNavi-NP, the predicted third candidate pathway belongs to one of the lysine (**8**) degradation pathways[55], and the seventh one is highly similar to an experimentally reconstructed pathway for glutaconate production[56], both of which are already existed in our training set, so it is not surprising that BioNavi-NP can predict those two pathways which have been constructed in *E. coli* for glutarate production[57,58]. The fourth candidate is not contained in our training set, and it takes a successive decarboxylation strategy from homoisocitrate (**9**), which is predicted to be originating from the basic building block pyruvate (**10**). Interestingly, Wang et al.[59] recently established a glutarate biosynthetic pathway from α-ketoglutarate by incorporation of a "+1" carbon chain extension and α-keto acid decarboxylation, which covers the fourth candidate pathway predicted herein. This case study is a typical example to show the capability of BioNavi-NP in exploring the biosynthetic pathways beyond the known biogenesis. Thus, it is useful for biosynthetic pathways redesign.

## Discussion

Herein, we proposed a practical retro-biosynthesis protocol for navigating biosynthetic pathways to complex NPs from simple building blocks. In contrast to rule-based models, BioNavi-NP is a fully data-driven model that is constructed based on curated biosynthetic and organic reaction data without the need for cumbersome or heuristic extraction of reaction templates. Comprehensive evaluations on internal and external test sets demonstrate that our method achieves a high success rate (90.2% over 368 internal molecules and 88% over 25 external molecules) when generating biosynthetic routes to complex NPs that recapitulate known pathways and building blocks, particularly in comparison to other methods at a comparable computational time. By combining an existing enzyme prediction tool, we further provide a user-friendly server that can not only predict biosynthetic pathways but also rank the biological feasibility of these pathways according to the estimated preference of species and enzymes. This is a valuable feature considering that template-free methods often predict new reactions outside of current knowledge. We have validated the potential of this webserver to predict native biosynthetic pathways as well as redesigning pathways for complex NPs, although we acknowledge that the pathway scoring function is not perfect due to the limited data availability and inherent limitations of the search algorithm. Therefore, extra modules that link the predicted intermediates and reactions to the known data sources were added to the webserver to offer annotation and facilitate the re-ranking of pathways. In this sense, we have improved the reproducibility of the reported biosynthesis routes and the enumeration ability of alternative biosynthetic pathways, provided an easy operational and visualizable webserver with some key options for users to balance accuracy and efficiency in practical use, and also supplemented multiple criteria for judging the rationality of the predicted pathways and building blocks. Future improvements can be made by providing further information, such as the promiscuity, fidelity, and diversity of enzyme catalysts, and introducing advanced enzyme prediction and design tools[60–62] to aid experimental biosynthesis.

There still remain certain challenges for the refinement of the BioNavi-NP. For example, at present, it could not correctly predict the biosynthetic starting materials for some complex NPs involving many building blocks or reaction steps, even though a much longer pathway could be found compared to some other methods (Table 2). Alternatively, a complete biosynthetic pathway of a complex structure may be able to be predicted by dividing it into multiple segments (by algorithm or manually) and

then merging each part. Additionally, intermediates may be incorrect or missing in some predictions. Even so, the results may also be practical to experimental test for many cases, since the missing parts can be inferred by the predicted steps (such as the dimethylallyl diphosphate in case 1 and 2, Supplementary Fig. 6), or they can be existed in other output pathways (e.g. the fourth and fifth candidate constitute the complete biosynthetic pathway of sterhirsutin J, Fig. 3b). In other words, to navigate the complicated biosynthetic pathways network of NPs, it is best to consider all of the candidate pathways in the network holistically; integration or correction may be required to design a highly-efficient biosynthetic pathway. Furthermore, a few predicted pathways tend to take shortcuts through small cofactor-type molecules or by-products. In fact, efforts have been made in data pre-processing to reduce the ambiguous role of generic compounds, but there are still a small fraction of anomalous data leading to unjustified shortcuts. Future work on integrating atom-mapping approaches[63,64] into the rule-free models will allow better mass-balance for the single-step retro-biosynthesis predictions.

In summary, this work combines transformer models and the Retro* search algorithm to develop a leading-edge biosynthesis navigator (BioNavi-NP), which can enumerate diverse biosynthetic pathways and trace natural products back to biologically-plausible building blocks. BioNavi-NP achieves a high success rate at generating biosynthetic routes and searching building blocks for NPs. Therefore, it is promising for native biogenesis analysis, biosynthetic pathway reconstruction, and rational design.

## Methods

**Data set**. MetaNetX[7] integrated a number of major resources of metabolites and biochemical reactions including MetaCyc[5], KEGG[6], The SEED[65], Rhea[66], BiGG[67] and so on. However, the information regarding reaction directionality and compartmentalization were generally disregarded, which are important to predict the precursors of the NPs. So we first extracted reactions directly from MetaCyc (version 23.5) pathway ontology and KEGG (accessed in January, 2021) pathway maps, where the "irrelevant" agents such as cofactors and minor substrates or products were removed. These reactions were excluded from MetaNetX (version 4.1) dataset, and following the previous work[34], we derived pairs of precursors and metabolites where the maximum common substructure exceeds 40% of the atoms of the metabolite for the rest of the reactions. Then, multi-product reactions were decomposed to multiple mono-product reactions with the same substrates and the cofactors were removed. Compounds containing Coenzyme A (CoA) play an important role in the biosynthesis of natural products especially polyketides and lipids. Although CoA is a complex structure that contains 84 heavy atoms, it is usually not involved in the reaction center, so the CoAs in the molecules were replaced by "*" in the SMILES format and restored when the results were output. Considering that the stereochemistry is often poorly annotated in the structure databases, we firstly checked the structures in our reaction dataset. The potential stereochemical centers exist in 15,372 out of 20,710 structures, and 10,909 (71.0%) of them are completely annotated with the stereochemistry, 12,763 (83.0%) are annotated with at least half of the stereochemistry. Thus the stereochemistry was preserved in our dataset. The resulting BioChem dataset consists of 33,710 unique pairs of precursors and metabolites, 1000 of which were selected as the test set and another 1000 as the validation set (see Supplementary Fig. 2). Additionally, in order to learn more syntactic rules for chemical reactions, we screened reactions with components similar to natural products from the organic chemistry dataset for model pre-training. Specifically, we used molecular ECFP4 fingerprints[68] to select the reactions with a maximum similarity of ≥0.8 (calculated by RDKit[69]) between any component (reactants and/or products) of each reaction in the USPTO database and the NPs from the DNP[1] database (version 27.2), resulting in 60,000 chemical reaction equations. The USPTO_NPL dataset was merged with the Bio-Chem dataset to train the single-step model.

To obtain a set of target natural products for the quantitative evaluation of a multi-step model that contains different scaffolds, structures from the BioChem dataset were clustered according to their biosynthetic building blocks. Then 368 target molecules with complete biosynthetic pathways (internal cases) were randomly extracted from the different clusters, which can be further divided into five broad categories (the Acetic Acid and malonic acid pathways, AA/MA; the mevalonic acid or methylerythritol phosphate pathway, MVA/MEP; the cinnamic acid or shikimic acid pathway, CA/SA; amino acids pathway, AAs; the Other pathway, including the hybrid pathway mentioned above). Moreover, 25 unseen natural products (external cases) whose biosynthetic pathways are unclear or not included in the data source were chosen to evaluate the generalization of our model.

**Computational models**. Given an input of a target molecule, our task is to recursively decompose the molecule into available biogenetic precursors following the biosynthesis scheme. In this study, a biogenetic reaction was described by a variable-length string containing one pair of SMILES notations representing the reactants and target compound (Supplementary Fig. 1a). Each reaction was split into a source sequence and target sequence for model training. For example, a biogenetic reaction for 2-Amino-5-Chlorophenol can be described as "Nc1ccc(Cl)cc1OC(=O)O≫Nc1ccc(Cl)cc1O", where the "Nc1ccc(Cl)cc1O" is the source sequence, the "Nc1ccc(Cl)cc1OC(=O)O" is the target sequence. The building block library has to be selected before the planning is started. Different from organic synthesis and metabolic engineering, we only assigned 40 building blocks (by default, extended and user-defined library are also available) from three major sources amino acids, organic acids and other molecules upstream the biosynthetic pathway of natural products (Supplementary Fig. 3), which have been proved to be the basic components of most of the natural products[3,4].

The single-step prediction was made by the transformer neural networks[30], which were built, trained and tested by Pytorch[70] and OpenNMT[71] framework. Our best-performance single-step models were trained for 48 h on four GPU (Nvidia 2080TI) on the training set, saving one checkpoint every 10,000 steps and averaging the last five checkpoints. Hyperparameters are selected based on the performance of the model on the validation set. A beam search procedure[72] was then used to infer multiple precursors candidates on the test set. We used the best performance model to infer the candidate sequences of precursors with a beamwidth of 10. As a result, the top10 candidate sequences ranked by total probability were retained. The training details can be seen Supplementary Method 1.

For multi-step planning, we adopt Retro*[39] (Supplementary Method 1 and Supplementary Fig. 1b) as the search engine to find high-quality synthetic routes efficiently. Retro* is a best-first search algorithm, which exploits neural priors to directly optimize for the quality of the solution. It translates the search as an AND-OR tree and learns a neural search bias with off-policy data. Specifically, the search tree $T$ is an AND-OR tree, with molecule node as 'OR' node and reaction node as 'AND' node. It starts the search tree $T$ with a single root compound node, which is the target compound $t$. At each iteration, it selects a node $u$ in the frontier of $T$ (denoted as $\mathcal{F}(T)$) according to the value function. Then it expands $u$ with the trained single-step transformer neural networks and grows $T$ with one AND-OR stump. Such expansion accumulates the corresponding cost information for single-step retrosynthesis. The cost of single-step retrosynthesis is reflected by the confidence score (the negative loglikelihood of this reaction under transformer model, smaller prediction cost means higher reaction probability) and the total cost is the sum of all steps scores along the pathway.

Each time a full pathway is found during the tree expansion, the pathway is returned, and an additional bonus is received by the node, to allow for biasing toward similar successful pathways. At the end of the search, the most visited pathway is returned (i.e. the best one), and all pathways are returned ranked in order of decreasing cost. When performing multi-step evaluation on the internal set, we set the beam size of the transformer, max route, iteration, max depth as 10, 5, 100 and 10, respectively. It is worth noting that increasing the width, depth of the search and the number of iterations will certainly improve the accuracy, but the time consumption will also increase exponentially.

**Reporting summary**. Further information on research design is available in the Nature Research Reporting Summary linked to this article.

## Data availability

The raw data is collected from KEGG (https://www.kegg.jp/), MeatCyc (https://metacyc.org/), and MetaNetX (https://www.metanetx.org/). The processed data that used to train and test the model is available at BioNavi-NP [http://biopathnavi.qmclab.com/data.html]. The raw output of external cases are also available at BioNavi-NP [http://biopathnavi.qmclab.com/case.html]. Source data are provided with this paper.

## Code availability

BioNavi-NP can be accessed freely as the Web server http://biopathnavi.qmclab.com/. The source code is available at Github [https://github.com/prokia/BioNavi-NP].

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

## Acknowledgements

This work was supported by the National Key R&D Program of China (2020YFB0204803), National Natural Science Foundation of China (21773313 and 61772566), Guangdong Key Field R&D Plan (2019B020228001 and 2018B010109006), Guangzhou S&T Research Plan (202007030010), and Sun Yat-sen University (20ykzd13). We thank the Guangzhou, Shenzhen Supercomputer Centers as well as Galixir for providing computational source. We thank Yong Liu and Jixian Zhang for helpful discussions.

## Author contributions

R.W. designed and supervised the whole research. S.Z., T.Z. and Y.Y. contributed concept and implementation. S.Z, T.Z. and C.L. contributed the development of webserver. B.C. ran some initial experiments. C.W.C. participated in the discussion and revision of the manuscript. All authors contributed to the interpretation of results. R.W., S.Z. and T.Z. wrote the manuscript. All authors reviewed and approved the final manuscript.

## Competing interests

S.Z. and C.L. are employees of Galixir, and C.W.C. is an advisor to Galixir during the course of this work. Other authors declare no competing interests.
