## [Peer Review File · Nature Communications]

Deep Learning Driven Biosynthetic Pathways Navigation for Natural Products with BioNavi-NPReviewers' Comments:

Reviewer #1:

Remarks to the Author:

General:

The authors present the adaptation of an existing retrosynthetic planning tool for retrobiosynthetic prediction of pathways towards natural products. They show that their purely machine-learning-based tool outperforms a current rule-based retrobiosynthesis algorithm. The predictions are impressive given the fact that a biochemically "blind" algorithm generates them. Still, it is not very clear how realistic the predictions are in a biological context. To go from a predicted transformation to a hypothesis of which enzyme could perform the predicted step, one would need a minimum of further information, such as an indicator of whether the reaction is already known or not, the reaction mechanism, cofactors involved, enzyme class, or similarity with known enzymes. Hence, it is not clear from the paper how the output of BioNavi-NP can be used for metabolic pathway engineering. This limiting point is raised by the authors themselves in the discussion but still is a significant drawback of the work.

Another problem of the pathway prediction is that there is no notion of mass balance: The number of atoms in the suggested building blocks is insufficient to account for the atoms in the parent structure. The interactive web service is straightforward to use and nicely displays the results. However, it would be helpful to add a description of the different parameters if the user would like to choose non-default parameters.

Regarding the language, we would suggest a professional editing service, in particular also for the abstract. The authors should ensure that the terminology is explained/defined at the first occurrence in the text.

In the discussion, authors state that the BioNavi-NP is a good navigator instead of a scorer (l. 274) and propose to view the whole pathway network generated by their approach rather than predict pathways based on it; therefore, the comparison with the RetroPathRL is in the end not entirely uniform, in this case, it maybe would be better to compare the pathway networks generated with BioNavi-NP with the pathway networks generated by other tools.

Overall, the understanding of the manuscript is complicated as the main text and the supplementary material is intertwined. The main text does not allow a comprehensive understanding of the method. While it is appropriate to keep the unessential details for the supplemental material, we would recommend explaining the essential parts of the method in the main text since the method is the main research result of this publication.

Moreover, the methods in supplementary material are described in a way that might be challenging to understand for the general scientific public, which is the target audience of the Nat Comm (especially paragraphs highlighted below as "this part is not well explained").

The platform:

- The platform is user-friendly. Compounds can be queried with their SMILES or InCHI. Then depending on the complexity of the molecule, it will take several minutes till the list of pathway searches be visualized.

The pathway visualization is interactive with a chain of molecules between queried compounds and the building blocks. Each edge in this chain is annotated with a score of how likely the transformation is. Capturing different transformations and ranking the best pathways only based on a machine learning algorithm is impressive. However, we are skeptical that the authors make a good case for adopting their platform in the pathway discovery field. In practice, the users need to rank alternative pathway solutions based on criteria such as the number of known reactions in the pathway, involved cofactors, enzyme information (prediction in case of novel transformation), etc. The authors would have to annotate their results with more details; otherwise, interpretation of the raw results and concluding is challenging.

Furthermore, linking the known transformations to the external resources (exp, KEGG, MetaCyc,...)

increases the reproducibility, consistency, and transparency of the results.

- When we tried to run another example in the platform, the browser returned an error. We admit that we did not have time to try again, which might be a temporary error. However, we draw attention to the fact that this initial experience, if persistent, will detract many users and decrease the impact of the platform.

Major:

Line 86 - 88: "However, these have not been applied to retro-biosynthesis planning. Overall, there is yet no practical computational tool for retro-biosynthesis analysis, due to the multi-step and multi-pass features of biosynthetic pathways for most NPs "

What are multi-pass features, and why are the problems they refer to for the retrobiosynthesis tools? How is the multi-step feature a problem for retrobiosynthesis tools ?

The re-cap on existing methods and past work is quite limited and misses an overview on different approaches. It is unclear how the presented work differs from previous work in the field of retrobiosynthesis, and therefore difficult to situate the proposed work within the various existing retrobiosynthesis methods. Also, since the authors compare their results to RetroPathRL, it would be good to explain the differences and similarities between the two methods in detail.

Line 94-96: "For a target NPs compound, its biosynthetic precursors are predicted by an ensemble of four Transformer neural networks that recursively propose one-step disconnections until plausible building blocks are identified. "

What are Transformer neural networks ? What are "one-step disconnections"?

Line 98: "Finally, the huge number of reaction pathways are ranked before the top-N pathways are visualized by an interactive website"

Details of ranking are essential, and here there is only reference to the supplementary material. Later in the discussion, the authors say that their tool is unsuitable for ranking instead of generating the network for the general overview.

Line 109: Where does the training data for the 31710 enzymatic reactions come from? Why are organic reactions also used for training? We see that they improve the model performance, but what is the rationale behind it or the hypothesis explaining why this works?

It would be nice to see some examples of reactions where the knowledge on organic chemistry actually helped to recover the reaction prediction.

Line 113: What is the "pretrained model without fine-tuning"? Is it a model only trained on organic reactions?

Line 120: How does the accuracy of single predictions compare to reported accuracies from other tools, e.g., RetroPathRL? Also, it would be helpful to know what "USPTO_NPL + BioChem "is only defined later in the text.

Line 137: Where does the pathway data come from that is used for validation?

Line 139: In the RetroPathRL paper, Koch et al. report that is using their best configuration, they find potential biosynthetic pathways for all 20 of their 20 pathways (100% success rates) in what they call their "golden data set," and they get a success rate of 83.6% using they their best configuration for a compilation of 152 successfully engineered metabolic pathways (LASER data set). This does not match well with the success rate of 52.7% that the authors report for RetropathRL. Where does this difference in success rates for the same tool come from?

For a fair comparison of the two methods, we suggest evaluating BioNavi-NP on the dataset used by Koch et al. The same also applies to the reported hit rates.

Line 177: Some of the 25 predicted pathways look pretty reasonable (e.g., case 13, 14), but the pathways predicted for the bigger molecules are somewhat unclear. For example, only three reaction steps are predicted for the complex molecule in case 24. Still, by looking at the structure, one can tell that its biosynthesis would require more than three reaction steps from common precursor molecules. For other cases, the predictions do not explain the provenance of all the atoms in the molecules (e.g., case 25).

Line 179-186: This part is difficult to understand: Which results are instructive, and what do they tell us? What do you mean by domain expertise?

Line 220: How is the "predicted cost of each step" calculated, what does it represent, and how shall one interpret it? Some explanation is given later (line 255), but it would be necessary for the reader to define it here.

Line 261: "without needs of cumbersome extraction of reaction templates or biased expert systems"- Why are expert systems biased?

Figure 1: Requires more details in the caption; it is unclear what Ensemble and Biochemical Transformer is.

Supplementary information:

Line 64: "The encoder layers are input with the source molecular embedding. For single-step retrosynthetic prediction, we adapted the Transformer architecture and iteratively transform it into a latent representation. After completing the encoder stage, each step in the decoding stage outputs a token based on the latent information I until the end token '</s>' is reached, indicating the completion of the output of the transformer decoder." - This part is not well explained

Line 88: "which exploits neural priors to directly optimize for the quality of the solution"- What is neural priors?

Line 118: MCTS not defined in supplementary, only in the manuscript (Monte Carlo tree search)

Line 144: "'irrelevant' agents such as cofactors and minor substrates or products were removed"- what are the minor substrates or products? What if they can affect the retrosynthesis planning as they can not be produced by the organism / can be toxic?

Line 145: "we derived pairs of precursors and metabolites where the common atoms exceed 40% of the atoms of the metabolite." - How the number of common atoms was calculated?

Line 154: "Additionally, the NP-like dataset containing 60000 reactions was organized from USPTO12 based on fingerprint similarity (threshold was 0.8) between components of reaction and NPs from the DNP database (version 27.2)." - This part is not well explained.

Minor:

Line 59: Typo in "Metohds"

Line 137: "are already existed" -> "exist"

Line 254: "Problem in "The options for highlight pathway [...]"

Line 293: Verb missing

Line 299: should be "challenging"

Reviewer #2:

Remarks to the Author:

This contribution reports on the development and evaluation of a software toolkit for the prediction of the biosynthetic pathways for natural products (NPs) and NP derivatives. In my opinion, this work, in its current form, does not meet the standards for publication in Nature Communications (or elsewhere) for the reasons pointed out below.

Firstly, the manuscript is difficult to read because much of the ESSENTIAL information on algorithms, datasets, etc. is either hidden somewhere in the Supp Inf or missing entirely. I understand that this is supposed to be a short communication but it is not very helpful if information ESSENTIAL for the basic understanding of the relevance and validity of a method is not included in the main text. Having read the manuscript more than once, I still have only a very vague idea about the scope and limitations of the method and, consequently, its relevance.

I am concerned that parts of the manuscript read more like a superficial advertisement than rock-solid science because many numbers (results) are presented to the reader often without the necessary scrutiny. For example, there seems to be not any analysis in to the directions of: What cases are covered by the model vs. when does the model fail (and why)? How does the model perform on NPs from different kingdoms? What NP space is covered by the training data? What is the applicability domain of the model? What is the (exact) structural and pathway relationship between the training and the test data?

It seems that there is no information included in the manuscript that would allow the reader to understand how challenging the presented test cases are (and, to some extent, also whether they are representative of the NP space). In fact, the presented information does not allow a clear conclusion as to whether some of the test data is part of the training data (e.g. the test cases reported on page 9).

The language is sometimes quite poor and many statements are unclear.

For time reasons I can only point out a few, specific examples of issues:

- The first sentence of the Abstract is symptomatic for the problems observed with this manuscript: "More than 300,000 natural products (NPs) are discovered while fewer than 30,000 validated NPs compounds involved in about 33,000 known enzyme catalytic reactions are validated, and the complete biosynthesis pathways are uncovered for most NPs."

The sentence is difficult to read and vague. What does "validated" mean? (apart from the fact that it is mentioned twice in the first sentence?) What does "verified" mean? "the complete biosynthesis pathways are uncovered for most NPs" reads weird: What does "complete" mean? Does it mean that there's no knowledge existing about this pathway at all, or are just parts of the pathway not known?

- L31: What does "verified" mean?

- TOC graphic: I don't find it informative nor intuitive – not sure what is depicted here

- L45: "NPs are produced by organisms from all kingdoms in nature". I don't understand this statement. Are there any organisms not producing any NPs?

- L46: Any reference for the cited "300,000 NPs" is missing.

- L46: "vast chemical space is reachable from dozens of simple building blocks: Any reference is missing (it doesn't help the reader if a reference for this is provided somewhere hidden in the Supp Inf)

- L80: "data-based similar pathway matching with known reactions" reads weird and difficult

- The figure captions are generally insufficient for a reader to make much sense out of the depictions without needed to look up details in the main text. For example, Figure 1a is non-intuitive and non-

informative, and the corresponding caption doesn't provide any of the much-needed information. What is the definition of a biosynthetic pathway?

- In the manuscript, virtually none of the ESSENTIAL information on the Transformer models and the data set is presented. It is impossible for a reader what data was worked on with what exact methods. For example, what is Retro*?

In my opinion the reader must not be forced to look up information that is ESSENTIAL to understand the overall idea of an experiment or approach in the Supp Inf.

- L108: accuracies are provided for the biosynthesis dataset, with reference to Table 1. However, only by comparing the numbers in this statement and in Table 1, a reader can guess which of the dataset the "biosynthesis dataset" is.

- L109: What is the definition of accuracy in this context? What data is the model even tested on?

- L110: What is USPTO_NPL?! Not introduced anywhere...

- L116: "This is in line with the fact that biosynthetic NPs and chemically synthetic compounds share some commonalities, but comprise two distinct structural spaces and distinct sets of reaction types, confirming that existing organic retrosynthesis tools cannot be directly used for biosynthesis prediction": It does not confirm this, it may suggest it. For a confirmation, the authors would actually need to test the existing organic retrosynthesis tools

- L117: The Authors should discuss the quality and comprehensiveness of the stereochemical information. These types of annotations are often of poor quality and incomplete.

- L122: "The performance is even comparable to those on predictions of organic retrosynthesis²⁴ (the top-10 accuracy of ~60%) , suggesting the power of deep learning-based method for the planning of biosynthetic processes." Does the cited work represent the state-of-the-art? How can the two studies be even compared? It seems that numbers are thrown at the reader without the necessary care and disclaimers

OK, so when checking the citation I read that there are accuracies reported up to 88.1% so I think that the description is not accurate and that seems not the right conclusion here. It actually shows that natural products' retro-biosynthesis is more challenging if the results are comparable.

- L127: "internal cases, see Supplementary Methods Section 3": Supplementary Methods Section 3 does not mention "internal" cases

- L128: "internal cases": What does this mean?

- L128: The authors state that they compared different computational methods. It is not clear what the relevance of these methods are, whether they represent the state of the art, and how they were selected.

- L137: where do these target NPs come from and what is their relationship with the training data? How do the models perform depending on their relationship of the target from the training NPs? There doesn't seem to be any real information about how well these models really work. There are just many numbers that a reader cannot learn much from

- L139: No information about why the RetroPathRL model was selected and whether it is applicable to the test data

- L166: It is not the goal of the model to aim at developing something

- L170: The authors test their models on 25 unseen NPs and find good performance. I'm sorry but I don't see any information on whether these compounds are easy or challenging to predict. I feel like being presented the unquestioned output of a black box

- L173: here and elsewhere: please comment on the chance probabilities

- L181: Again, at least I don't have a clue why this particular model/dataset was used at this point.

- L191: numbers are provided without any information/units

- Caption of figure 2 is not well readable (zig-zag flow)

- L216: what's a CASP tool?

- L221: why have these compounds been selected for case studies?! What is their relationship to the training data. The test could be absolutely trivial

- L251: the names of the options mentioned here are not consistent with the names shown in figure 3

- L259: "In contrast to earlier works" could the authors please enlighten the reader what earlier works they are referring to? Is this statement referring to the most advanced "earlier works", because there may always be "bad" earlier works?

- L261: "biased" seems to be used in a negative way, but bias may actually be positive in cases where the data are sparse
- L307: "BioNavi-NP outperforms other computational tools" Many of the tools referred to in this work are known to not be applicable to biosynthesis prediction hence the statement likely is misleading
- L322: "The source code is available from the corresponding author upon reasonable request" This could mean anything. The authors should provide the source on defined terms via a public code repository
- L337: Please explain what this company is doing otherwise the information on the conflict of interest is not interpretable
- Table 1: inconsistent use of N and n. And why are there no absolute numbers presented for Table 1 (they are reported in Table 2)?
- L252-253, "Herein the outputs are redrawn to be clear (the raw output provided in Supplementary Figure 6-S30)": which are the corresponding figures for the two examples in Figure 3b?
- L270: I don't understand why the Authors spend so many words on possible future directions while the manuscript falls short on many essential pieces of information to understand the current model
- L275-276: "...missing intermediates/building blocks if considering the fifth and sixth candidates separately": the sixth candidate pathway is not marked and it is hard to understand what the authors are trying to say
- L290-292: "When BioNavi-NP finds an unreasonable pathway in the first prediction, it is worth a second prediction to using a well-validated or user-confident intermediate as the target molecule." Does "the first prediction" mean the first candidate pathway?
- Are there duplicates between USPTO_NPL and BioChem? The description of " the NP-like dataset containing 60000 reactions was organized from USPTO12 based on fingerprint similarity (threshold was 0.8) between components of reaction and NPs from the DNP database" in the supporting information is not clear what fingerprint was used and what exactly was kept. What about the similarity of 1.0?

Reviewer #3:

Remarks to the Author:

Title: BioNavi-NP: Biosynthesis Navigator for Natural Products

Authors: Shuangjia Zheng, Tao Zeng, Chengtao Li, Binghong Chen, Connor W. Coley, Yuedong Yang, Ruibo Wu

Recommendation: Accept with minor revisions

Summary: An ensemble of four transformer neural networks supplemented by a multi-step Retro* algorithm is proposed as a method of hypothesizing biosynthesis pathways for natural products (NPs) for which these pathways are currently unknown. The four transformer neural networks predict a single biosynthesis step, hypothesizing the building blocks for the NP in question, with the Retro* strategy filling in the intermediate step details. The effects of training data and consideration of chirality is investigated. A few case studies are conducted. This work is linked to a public website which can be used by researchers to investigate potential biosynthesis pathways for organisms of interest.

General reviewer comments: Overall, this work presents great opportunities for the production of NPs, as NPs cannot be biomanufactured reliably without understanding their synthesis pathways. It appears the neural networks are more successful at predicting the building blocks of the NP than the Retro* strategy is at predicting the intermediate steps.

Major Concerns:

1) Overall: Biosynthesis of NPs, particularly those with peculiar products, are often restricted to a specific species, genus, or other biological subgroup. Synthesis pathways, even for the same or similar compounds, can vary greatly between dissimilar organisms, yet there is no discussion of the considerations of species or phylogeny of an organism when predicting biosynthesis pathways. The lack of consideration for the species in question for biosynthesis is, I believe, the greatest weakness of this work as presented. While adding this would be beyond the scope of the current work, a discussion of considering this as a future research direction would be appropriate.

Minor Concerns:

1) TOC graphic: While the reference to Legos as "building blocks" of NP is clear, though most of the rest of your graphic is unclear. For instance, the transparency of the Lego blocks does not seem to have a purpose, neither does the compass underneath the desired NP. Further, it is unclear why the arrows move from the natural product to the basic building blocks, if it is desired to elucidate the NP biosynthesis pathway (arrows going in the other direction might make more sense). Overall, I think more words or labels are necessary in this figure to accurately convey meaning.

2) Discussion: While implementing this suggestion is beyond the scope of the current study, combining this tool with the prediction of the enzyme which could catalyze the predicted steps could be quite exciting and useful, as suggested in the discussion. This could also provide species- or genus- specific checks on the biological feasibility of pathways. Pathway likelihood could then be modulated by the enzymatic catalytic capabilities, as suggested in the manuscript's discussion. If this could be added onto this BioNavi-NP tool, this could be a transformative and widely used tool. Would it be possible to incorporate model tools like or pfam or machine learning-based tools such as AlphaFold as a way to predict whether or not biocatalysis of a predicted reaction step is feasible? If this is feasible, consider including this in the discussion of enzymes in the discussion section, as this section is currently missing a discussion of the genome of the organism synthesizing the NPs in question, rather this tool just considers a general biosynthetic pathway.

3) Machine learning algorithm used. While in the manuscript considerations like chirality and training data for effectiveness of the BioNavi-NP tool, no discussion has been given to the machine learning algorithm used. Is a transformer neural network a standard for this type of application? Why was this particular method chosen over other machine learning techniques? How was the number of inputs or layers for these networks chosen? Further, it's not explained why an ensemble of four neural networks is used. Is this related to the four major classes of NPs discussed in the introduction?

Concerns of grammar, clarity, spelling, and similar:

1) Abstract. Throughout the abstract, errors in grammar and clarity are present, sometimes presenting the opposite meaning from what I believe is intended. Consider this:

a. Abstract. "More than 300,000 natural products (NPs) are discovered while fewer than 30,000 validated NPs compounds involved in about 33,000 known enzyme catalytic reactions are validated, and the complete biosynthesis pathways are uncovered for most NPs".

i. The first phrase ("More than 300,000 natural products (NPs) are discovered while fewer than 30,000 validated Ps compounds involved in about 33,000 known enzyme catalytic reactions are validated") is awkward to read. It may be best reformulated as "More than 300,000 natural products (NPs) have been discovered while fewer than 30,000 validated NP compounds involved in about 33,000 known enzyme-catalyzed reactions are validated".

ii. The second phrase ("and the complete biosynthesis pathways are uncovered for most NPs") means that the complete biosynthesis pathways are known for most NPs rather than, as you intend, to state that they are unknown. This would make your tool unnecessary. This could be phrased as "and the complete biosynthesis pathways are still unknown for most NPs."

b. This is just one example of unclear wording, sentence structure, or grammar. It is highly recommended that this abstract be thoroughly proofread and edited. On the contrary, the body of the paper appears well-written and clear, so perhaps the writer of the body can proofread/edit the abstract.

2) Page 7, line 174: plurality issue, original: "...and at least one building blocks were correctly identified...", corrected: "...and at least one building block was correctly identified...".

3) Discussion section: The paragraphs stating the opportunities for improvement for BioNavi-NP should be labeled "(a)", "(b)", and so forth. Rather than the last sentence at line 269 on page 11, which should be reformatted as an introductory sentence for the next few paragraphs, the paragraphs do not need labels. Then each first sentence of these improvement paragraphs should be reworded as introductory sentences for the paragraphs.

4) Page 12 line 297: Original: "catalysis make it challenging to enzyme assignment", correction: "catalysis make it challenging to assign enzymes" or similar.

5) Please have another round of proofreading and editing for the paper to catch any issues I have not explicitly identified here.

Reviewer #4:

None

Reviewer #5:

None

Reviewer #6:

None

Summary response to reviewers:

- We would like to thank all Reviewers' detailed interest in our work and the constructive comments that helped us to improve our results. We have now **revised the whole manuscript and added more descriptions** about the methods, dataset and results in the new version.
- We have also **expanded our single-step as well as multi-step benchmark tests**, accordingly, Table 1 was updated and Supplementary Table S1 was newly added (see below).

Table 1 | Performance of single-step models by different training strategies.

Training strategy*	top-N accuracy (%), N =			
	1	3	5	10
USPTO_NPL	0	0	0	0
BioChem (w/o chirality)	7.6	11.1	13.9	16.3
BioChem	10.6	20.1	24.5	27.8
BioChem + USPTO_NPL	17.2	30.2	41.9	48.2
BioChem + USPTO_NPL (ensemble)	21.7	42.1	52.4	60.6
BioChem + USPTO_NPL (seq2seq)	10.9	21.3	30.8	37.1
RetroPathRL	20.6	30.5	36.8	42.1

Supplementary Table 1. Compare of BioNavi-NP and RetroPathRL

	LASER dataset (152)		Golden dataset (20)	
	Success rate(%)	Time (h)	Hit rate of pathway(%)	Time (h)
RetroPath 2.0	77.6	-	55	-
RetroPathRL	83.6	105	75	8
BioNavi-NP	94.7	12	65	1.5

- For the webserver, the known reactions (linking to MetaNetX¹) and compounds (linking to PubChem²) have been annotated to the outputs. More importantly, an enzyme selection tool (Selenzyme³) has been integrated in our server to address reviewers' suggestions in terms of enzyme and species. The enzymes and corresponding scores can be obtained by accessing Selenzyme. The scores were mainly determined by the reaction similarity and the taxonomic distance between the organism of enzyme and user-defined organism, which can be used to re-rank the pathways according to the specific organism. We think these will make our tool more feasible and useful. Herein an example for illustration:

Supplementary Figure 31. The result panel of BioNavi-NP, which consists of the interactive pathway network and the summary table, where the pathways can be re-ranked by several items. Selenzyme was accessed to search enzymes and score for specific reactions.

Point-by-point response to reviewers:

Reviewer #1 (Expertise: Retrobiosynthesis, metabolic engineering):

General:

The authors present the adaptation of an existing retrosynthetic planning tool for retrobiosynthetic prediction of pathways towards natural products. They show that their purely machine-learning-based tool outperforms a current rule-based retrobiosynthesis algorithm. The predictions are impressive given the fact that a biochemically “blind” algorithm generates them. Still, it is not very clear how realistic the predictions are in a biological context. To go from a predicted transformation to a hypothesis of which enzyme could perform the predicted step, one would need a minimum of further information, such as an indicator of whether the reaction is already known or not, the reaction mechanism, cofactors involved, enzyme class, or similarity with known enzymes. Hence, it is not clear from the paper how the output of BioNavi-NP can be used for metabolic pathway engineering. This limiting point is raised by the authors themselves in the discussion but still is a significant drawback of the work.

- We agree with the reviewer and have added more annotations to the predicted pathways, please see details in the above summary response.

Another problem of the pathway prediction is that there is no notion of mass balance: The number of atoms in the suggested building blocks is insufficient to account for the atoms in the parent structure.

- BioNavi-NP aims to predict the biotransformation route from pre-defined building blocks to target natural product, so only the main components of the reactions were kept in the training set. And this “cleaning” process also stems from the intrinsic drawback of the neural network approach to SMILES, where previous work also performed the same cleaning operation on organic reactions during training, to avoiding too many by-products affecting the syntax of SMILES and thus resulting in illegitimate SMILES. Such strategy has also been widely used in other template-free retrosynthesis prediction^{4,5}. At present we do not have a better solution, but by annotating the output with known reactions and compounds and enzymes, users can conveniently justify the feasibility of the prediction pathway.

The interactive web service is straightforward to use and nicely displays the results. However, it would be helpful to add a description of the different parameters if the user would like to choose non-default parameters.

- Thank you for the suggestion. We have added the descriptions to make the server more user-friendly (see below), and they have also been added to the Supplementary Information.

Regarding the language, we would suggest a professional editing service, in particular also for the abstract. The authors should ensure that the terminology is explained/defined at the first occurrence in the text.

➤ Thank you for the suggestion and we have revised the whole manuscript accordingly.

In the discussion, authors state that the BioNavi-NP is a good navigator instead of a scorer (l. 274) and propose to view the whole pathway network generated by their approach rather than predict pathways based on it; therefore, the comparison with the RetroPathRL is in the end not entirely uniform, in this case, it maybe would be better to compare the pathway networks generated with BioNavi-NP with the pathway networks generated by other tools.

➤ We showcase the whole pathway network because we found that the ranking ability of our model seems not ideal for some cases, so all of the output pathways are recommended to be taken into consideration. And as described in the summary response, we added more annotations and enzyme selection module to help re-rank the output pathways based on users' knowledge. Since both BioNavi-NP and RetroPathRL were developed to make retro-biosynthesis predictions, we compared these two methods in four aspects (success rate, hit rate of building block/pathway, and longest length of pathway) highly related to the pathways, as summarized in Table 2.

Overall, the understanding of the manuscript is complicated as the main text and the supplementary material is intertwined. The main text does not allow a comprehensive understanding of the method. While it is appropriate to keep the unessential details for the supplemental material, we would recommend explaining the essential parts of the method in the main text since the method is the main research result of this publication.

Moreover, the methods in supplementary material are described in a way that might be challenging to understand for the general scientific public, which is the target audience of the Nat Comm (especially paragraphs highlighted below as “this part is not well explained”).

➤ Thank you for the very helpful suggestions, and we have re-organized the manuscript and added more details and give more clear description about the methods in the Methods section at the end of the revised manuscript.

The platform:

- The platform is user-friendly. Compounds can be queried with their SMILES or InCHI. Then depending on the complexity of the molecule, it will take several minutes till the list of pathway searches be visualized.

The pathway visualization is interactive with a chain of molecules between queried compounds and the building blocks. Each edge in this chain is annotated with a score of how likely the transformation is. Capturing different transformations and ranking the best pathways only based on a machine learning algorithm is impressive. However, we are skeptical that the authors make a good case for adopting their platform in the pathway discovery field. In practice, the users need to rank alternative pathway solutions based on criteria such as the number of known reactions in the pathway, involved cofactors, enzyme information (prediction in case of novel transformation), etc.

The authors would have to annotate their results with more details; otherwise, interpretation of the raw results and concluding is challenging.

Furthermore, linking the known transformations to the external resources (exp, KEGG, MetaCYC,...) increases the reproducibility, consistency, and transparency of the results.

➤ We appreciate the review's positive comments and we have improved the result annotations for ranking as suggested. Please see details in the summary response.

- When we tried to run another example in the platform, the browser returned an error. We admit that we did not have time to try again, which might be a temporary error. However, we draw attention to the fact that this initial experience, if persistent, will detract many users and decrease the impact of the platform.

➤ Thank you for the comment, we have debugged the webserver.

Major:

1. Line 86 - 88: "However, these have not been applied to retro-biosynthesis planning. Overall, there is yet no practical computational tool for retro-biosynthesis analysis, due to the multi-step and multi-pass features of biosynthetic pathways for most NPs "What are multi-pass features, and why are the problems they refer to for the retrobiosynthesis tools? How is the multi-step feature a problem for retrobiosynthesis tools? The re-cap on existing methods and past work is quite limited and misses an overview on different approaches. It is unclear how the presented work differs from previous work in the field of retrobiosynthesis, and therefore difficult to situate the proposed work within the various existing retrobiosynthesis methods. Also, since the authors compare their results to RetroPathRL, it would be good to explain the differences and similarities between the two methods in detail.

➤ Multi-pass means that in the biosynthesis of natural products (especially for the early stage of the pathway), there exist more than one biosynthetic route and they are often intertwined to form networks. So the "right" pathway is hard to define and be identified or evaluated. While for the multi-step, the search algorithm plays a key role in retrosynthesis. All of the retrosynthesis tools were based on the single-step prediction, and generally speaking, more than one outputs will be preserved in the single-step prediction and used for the next iteration. So as the number of steps increases, the outputs increase exponentially, which is too time-consuming to continue the iteration and rank the outputs. And the efficient and reliable search algorithm will help to select most likely "correct" outputs to continue iteration.

To be clear, we have rewrite this part and given more detailed description for the existing methods for single- and multi-step retrosynthesis prediction, and distinguishing our work from RetroPathRL.

2. Line 94-96: "For a target NPs compound, its biosynthetic precursors are predicted by an ensemble of four Transformer neural networks that recursively propose one-step disconnections until plausible building blocks are identified. "What are Transformer neural networks ? What are "one-step disconnections"?"

➤ Transformer neural network is a deep learning model that translates a string of text to another

one, here it was used to translate the “product” to “substrate”, both of which are represented by SMILESs. We named this task ‘one-step disconnections’ also known as single-step prediction. The original words may be confusing, so we have revised them.

3. Line 98: “Finally, the huge number of reaction pathways are ranked before the top-N pathways are visualized by an interactive website” Details of ranking are essential, and here there is only reference to the supplementary material. Later in the discussion, the authors say that their tool is unsuitable for ranking instead of generating the network for the general overview.
 - The ranking processes were performed based on the single-step cost (the negative loglikelihood of this reaction under Transformer model) and pathway length. This is not a contradiction. As the ranking process did not involve the domain knowledge but rely on the algorithms only, we believe that the entire pathway should be made available to users for further evaluation. We have added more details to describe the “cost” in the Method section.
4. Line 109: Where does the training data for the 31710 enzymatic reactions come from? Why are organic reactions also used for training? We see that they improve the model performance, but what is the rationale behind it or the hypothesis explaining why this works? It would be nice to see some examples of reactions where the knowledge on organic chemistry actually helped to recover the reaction prediction.
 - The data set (BioChem) were collected from the MetaCyc, KEGG and MetaNetX, and then split into training (31710), validation (1000) and test (1000) set. The rationale behind the use of organic reactions is motivated by transfer learning, where a model is trained on a task with abundant data and simultaneously or subsequently fine-tuned on another task with less data available. Such a method can help the model learn the general patterns and has recently led to significant advancements in the field of Natural Language Processing and molecular modeling⁶. We have add some sentences to clarify all these points in “Single-step evaluation” of Results section.
5. Line 113: What is the “pretrained model without fine-tuning”? Is it a model only trained on organic reactions?
 - Yes. We have revised the sentence to “*the model only trained on NP-like organic reactions (USPTO_NPL) does not make any...* ”.
6. Line 120: How does the accuracy of single predictions compare to reported accuracies from other tools, e.g., RetroPathRL? Also, it would be helpful to know what “USPTO_NPL + BioChem “is only defined later in the text.
 - Good point. We have extended Table 1 to compare with the single-step performance of RetroPathRL and recurrent neural network (seq2seq). The results showed that our model outperform RetroPathRL and seq2seq with a top-10 accuracy of 60.6 versus 42.1 and 37.1, respectively. These results are consistent to previous similar models in organic reactions.^{4,7} All model abbreviations are explained in the revised text.

7. Line 137: Where does the pathway data come from that is used for validation?
 - The internal set were randomly selected from the BioChem data set and the external set was collected from the literature. We have added more description in the Methods section.

8. Line 139: In the RetroPathRL paper, Koch et al. report that is using their best configuration, they find potential biosynthetic pathways for all 20 of their 20 pathways (100% success rates) in what they call their “golden data set,” and they get a success rate of 83.6% using they their best configuration for a compilation of 152 successfully engineered metabolic pathways (LASER data set). This does not match well with the success rate of 52.7% that the authors report for RetroPathRL. Where does this difference in success rates for the same tool come from? For a fair comparison of the two methods, we suggest evaluating BioNavi-NP on the dataset used by Koch et al. The same also applies to the reported hit rates.
 - Done as suggested. We have evaluated our method on Koch et al.’s golden data set and LASER data set in Supplementary Table 1. For the golden set, our method also found potential pathways for all cases (100% success rate) and the hit rate of pathway was 65%, for comparison, RetroPathRL was 75% for hit rate of pathway. Regarding to the hit rates are much higher than those reported in Table 2, it is mainly due to the structures in golden set are relatively simple than those in our internal test set, which are easier to make predictions. It suggests that BioNavi-NP performed better (26.1%) than RetroPathRL (4.1%) for those more complex target NPs. For the LASER dataset, the success rate of BioNavi-NP (94.7%) outperforms RetroPathRL (83.6%), again proving that our models can achieve better performance when the query structures are more complex.

9. Line 177: Some of the 25 predicted pathways look pretty reasonable (e.g., case 13, 14), but the pathways predicted for the bigger molecules are somewhat unclear. For example, only three reaction steps are predicted for the complex molecule in case 24. Still, by looking at the structure, one can tell that its biosynthesis would require more than three reaction steps from common precursor molecules. For other cases, the predictions do not explain the provenance of all the atoms in the molecules (e.g., case 25).
 - As discussed in “(a) Intermediates may be incorrect or missing in some predictions.” part of Discussions section, for some complex structures with multiple building blocks, some of the intermediates and branches can be missing or even the results can be unreasonable. This can be due to the drawback of the neural network approach to SMILES as stated above and less training data relative to the target structures. While retro-biosynthesis for natural products especially the complex structures is hard and our model have achieved so far the best performance compared to the most recent rule-based models. Alternatively, integrating all of the output pathways or submit another job with a well-validated intermediate according prior knowledge were proved to be helpful in some cases.

10. Line 179-186: This part is difficult to understand: Which results are instructive, and what do they tell us? What do you mean by domain expertise?
 - In this work 40 structures that account for the origins of most NPs were selected as the building blocks (core library). Actually, many compounds in the early stage of the

biosynthetic pathways including the well-known precursor metabolites (such as glycerate, adenine and so on) can be also treated as the building blocks since the routes from core library to these metabolites can be clear or these metabolites lie in the cytosol compartment of some microbial strain. So the biosynthetic pathways from these metabolites to the target structure are also instructive. Now more discussion have been provided in “External testing” part of Results section.

- Here the domain expertise means the prior knowledge, at times one may deduce that a natural products could be derived from one or several building blocks so only the specific ones are required in the building blocks list. This will filter the “wrong” pathways in the prediction process and increase the “accuracy”, which can be proved by the increase of hit rate of both building block and pathway in the BioNavi-NP_UDB model in Table 2.

11. Line 220: How is the “predicted cost of each step” calculated, what does it represent, and how shall one interpret it? Some explanation is given later (line 255), but it would be necessary for the reader to define it here.

- The cost is the negative loglikelihood of this reaction under Transformer models, which can be interpreted as the probability that the molecule will be predicted. We have added the explanation in Methods section.

12. Line 261: “without needs of cumbersome extraction of reaction templates or biased expert systems”- Why are expert systems biased?

- The word ‘biased’ here refers to the fact that experts will have different familiarity with different pathways, leading them to predict the reactions they are more familiar with. While as reviewer #2 reminded that this can be also positive in some cases, so we remove it in the revised manuscript.

13. Figure 1: Requires more details in the caption; it is unclear what Ensemble and Biochemical Transformer is.

- Captions have been complemented in the revised manuscript. And we renewed the Figure 1 with additional notation.

Supplementary information:

14. Line 64: “The encoder layers are input with the source molecular embedding. For single-step retrosynthetic prediction, we adapted the Transformer architecture and iteratively transform it into a latent representation. After completing the encoder stage, each step in the decoding stage outputs a token based on the latent information l until the end token ‘ ϵ ’ is reached, indicating the completion of the output of the transformer decoder.” - This part is not well explained

- We revised this paragraph to “*The encoder layers are input with the source molecular SMILES sequence and transform it into a latent representation. The decoder combines the latent output from the encoder and the decoder's preceding output to predict the target molecular SMILES sequence. The training goal is to minimize the gap between predicted*

molecular SMILES sequence and target molecular SMILES sequence so that the model can finally infer accurate precursors.”

15. Line 88: “which exploits neural priors to directly optimize for the quality of the solution”- What is neural priors?

- The neural priors refer to the O_t and O_m functions, where we used them to guide the searching and optimization of the solutions

16. Line 118: MCTS not defined in supplementary, only in the manuscript (Monte Carlo tree search)

- We have added the full name before the abbreviation.

17. Line 144: “‘irrelevant’ agents such as cofactors and minor substrates or products were removed”- what are the minor substrates or products? What if they can affect the retrosynthesis planning as they can not be produced by the organism / can be toxic?

- Minor substrate refers to compounds such as water, oxygen and so on. As described above, this “cleaning” process stems from the drawback of the neural network approach to SMILES, and we do not have a better solution. Herein, the feasibility of the prediction pathway are further estimated by annotating the output with known reactions and compounds and introducing Selenzyme.

18. Line 145: “we derived pairs of precursors and metabolites where the common atoms exceed 40% of the atoms of the metabolite.” – How the number of common atoms was calculated?

- Following Litsa’s work⁵, we used RDKit⁸ Maximum Common Substructure match function to find common atoms between precursors and metabolites. This part has been moved to “Data set” part of Methods section in main text, more details are given there.

19. Line 154: “Additionally, the NP-like dataset containing 60000 reactions was organized from USPTO12 based on fingerprint similarity (threshold was 0.8) between components of reaction and NPs from the DNP database (version 27.2).” – This part is not well explained.

- Thanks for pointing out, we revised the sentence to “*Additionally, in order to learn more syntactic rules for chemical reactions, we screened reactions with components similar to natural products from the organic chemistry dataset for model pre-training. Specifically, we used molecular ECFP4 fingerprints to select the reactions with a maximum similarity of >0.8 between any component (reactants and/or products) of each reaction in the USPTO database and the NPs from the DNP database (version 27.2), resulting in 60,000 chemical reaction equations.*”, and moved from SI to main text in Methods section.

Minor:

Line 59: Typo in “Metohds”

Line 137: “are already existed” -> “exist”

Line 254: “Problem in “The options for highlight pathway [...]”

Line 293: Verb missing

Line 299: should be “challenging

- Thank you. We have now fixed all these typos.

Reviewer #2 (Expertise: computational chemistry, drug design):

This contribution reports on the development and evaluation of a software toolkit for the prediction of the biosynthetic pathways for natural products (NPs) and NP derivatives. In my opinion, this work, in its current form, does not meet the standards for publication in Nature Communications (or elsewhere) for the reasons pointed out below.

Firstly, the manuscript is difficult to read because much of the ESSENTIAL information on algorithms, datasets, etc. is either hidden somewhere in the Supp Inf or missing entirely. I understand that this is supposed to be a short communication but it is not very helpful if information ESSENTIAL for the basic understanding of the relevance and validity of a method is not included in the main text. Having read the manuscript more than once, I still have only a very vague idea about the scope and limitations of the method and, consequently, its relevance. I am concerned that parts of the manuscript read more like a superficial advertisement than rock-solid science because many numbers (results) are presented to the reader often without the necessary scrutiny. For example, there seems to be not any analysis in to the directions of: What cases are covered by the model vs. when does the model fail (and why)? How does the model perform on NPs from different kingdoms? What NP space is covered by the training data? What is the applicability domain of the model? What is the (exact) structural and pathway relationship between the training and the test data?

It seems that there is no information included in the manuscript that would allow the reader to understand how challenging the presented test cases are (and, to some extent, also whether they are representative of the NP space). In fact, the presented information does not allow a clear conclusion as to whether some of the test data is part of the training data (e.g. the test cases reported on page 9).

- Thank you for the important comments, and we have re-organized the manuscript with more details have been added to the main text.

The language is sometimes quite poor and many statements are unclear.

For time reasons I can only point out a few, specific examples of issues:

- The first sentence of the Abstract is symptomatic for the problems observed with this manuscript: “More than 300,000 natural products (NPs) are discovered while fewer than 30,000 validated NPs compounds involved in about 33,000 known enzyme catalytic reactions are validated, and the complete biosynthesis pathways are uncovered for most NPs.”

The sentence is difficult to read and vague. What does “validated” mean? (apart from the fact that it is mentioned twice in the first sentence?) What does “verified” mean? “the complete biosynthesis pathways are uncovered for most NPs” reads weird: What does “complete” mean?

Does it mean that there's no knowledge existing about this pathway at all, or are just parts of the pathway not known?

➤ We thank the reviewer for pointing out the problems. The sentence has been corrected and the “complete” means that all transformation steps from the building blocks to target structure have been characterized.

- L31: What does “verified” mean?

➤ It means the experimentally verified reactions.

- TOC graphic: I don't find it informative nor intuitive – not sure what is depicted here

➤ We have revised the TOC and added more details.

- L45: “NPs are produced by organisms from all kingdoms in nature”. I don't understand this statement. Are there any organisms not producing any NPs?

➤ We have deleted this sentence.

- L46: Any reference for the cited “300,000 NPs” is missing.

➤ The data comes from the number of items of Dictionary of Natural Products and Super Natural II. We have added the references accordingly.

- L46: “vast chemical space is reachable from dozens of simple building blocks: Any reference is missing (it doesn't help the reader if a reference for this is provided somewhere hidden in the Supp Inf)

➤ We have added the references in the revised manuscript.

- L80: “data-based similar pathway matching with known reactions” reads weird and difficult

➤ The statement has been corrected as follows: *‘organizing pathways by similarity matching with known reactions.....’*

- The figure captions are generally insufficient for a reader to make much sense out of the depictions without needed to look up details in the main text. For example, Figure 1a is non-intuitive and non-informative, and the corresponding caption doesn't provide any of the much-needed information. What is the definition of a biosynthetic pathway?

➤ More details have been added to the figure captions. The biosynthetic pathway in this panel represents the reaction network of the collected biochemical reactions.

- In the manuscript, virtually none of the ESSENTIAL information on the Transformer models and the data set is presented. It is impossible for a reader what data was worked on with what exact methods. For example, what is Retro*?

In my opinion the reader must not be forced to look up information that is ESSENTIAL to understand the overall idea of an experiment or approach in the Supp Inf.

➤ Thank you for the very constructive suggestion. More details have been added or moved from SI to the Methods section.

- L108: accuracies are provided for the biosynthesis dataset, with reference to Table 1. However, only by comparing the numbers in this statement and in Table 1, a reader can guess which of the dataset the “biosynthesis dataset” is.

➤ Thank you for point this out. “biosynthesis dataset” is the BioChem dataset and we have corrected it.

- L109: What is the definition of accuracy in this context? What data is the model even tested on?

➤ We collected 33710 reactions from the biochemical reaction databases (BioChem dataset) and randomly split them into training (31710), validation (1000) and test (1000) set. The product of a reaction was input into the model and the potential substrates would be output. The accuracy is the ratio of the reactions that the model predicted the correct substrate in the test set. More details have been added or revised in the “Single-step evaluation” part of Results section and the Methods section.

- L110: What is USPTO_NPL?! Not introduced anywhere...

➤ USPTO_NPL means the dataset of organic reactions with components similar to the NPs. All of the models have been explained more detailed in the Methods section now.

- L116: “This is in line with the fact that biosynthetic NPs and chemically synthetic compounds share some commonalities, but comprise two distinct structural spaces and distinct sets of reaction types, confirming that existing organic retrosynthesis tools cannot be directly used for biosynthesis prediction”: It does not confirm this, it may suggest it. For a confirmation, the authors would actually need to test the existing organic retrosynthesis tools

➤ We agree with you. This statement has been removed.

- L117: The Authors should discuss the quality and comprehensiveness of the stereochemical information. These types of annotations are often of poor quality and incomplete.

➤ Thank you for your suggestions. We have checked our BioChem dataset and found that the potential stereochemical centers existed in 15372 out of 20710 structures. 10909 (71.0%) of them have been completely annotated and 12763 (83.0%) have been annotated at least half of the stereochemical centers. These information were mentioned in Methods section. By the way, our result shown in Table 1 indicated that considering the stereochemistry will improve the performance of model.

- L122: “The performance is even comparable to those on predictions of organic retrosynthesis²⁴ (the top-10 accuracy of ~60%), suggesting the power of deep learning-based method for the planning of biosynthetic processes.” Does the cited work represent the state-of-the-art? How can the two studies be even compared? It seems that numbers are thrown at the reader without the necessary care and disclaimers

OK, so when checking the citation I read that there are accuracies reported up to 88.1% so I think that the description is not accurate and that seems not the right conclusion here. It actually shows that natural products' retro-biosynthesis is more challenging if the results are comparable.

- Strictly speaking, the cited work is the state-of-the-art of the *template-free methods* for retro-synthesis. We compare with these two studies because they both treated the retrosynthesis as language translation and used similar model (Transformer neural network) to perform the task. To be a relative fair comparison, we compared BioNavi-NP to two new methods (RetroPathRL and recurrent neural network (seq2seq)) in Table 1, and revised the statements as follows:

“Training the model with 60K organic reactions involving natural product-like compounds for augmentation (BioChem + USPTO_NPL) improves both accuracies to 17.2% and 48.2%, respectively, which has exceeded the top-10 accuracy of the state-of-the-art rule-based biosynthesis model, RetropathRL, with top-10 accuracy being 42.1%”

- L127: “internal cases, see Supplementary Methods Section 3”: Supplementary Methods Section 3 does not mention “internal” cases

- This has been fixed.

- L128: “internal cases”: What does this mean?

- It means the 368 cases randomly selected from the BioChem dataset.

- L128: The authors state that they compared different computational methods. It is not clear what the relevance of these methods are, whether they represent the state of the art, and how they were selected.

- We have added more description to compare different computational methods in the Introduction section, describing the existing works and the selected baselines for single- and multi-step retrosynthesis prediction, and showing the motivation of our work.

- L137: where do these target NPs come from and what is their relationship with the training data? How do the models perform depending on their relationship of the target from the training NPs? There doesn't seem to be any real information about how well these models really work. There are just many numbers that a reader cannot learn much from

- The 368 internal cases were randomly selected from the BioChem dataset and the external cases were collected from the recent literature that were independent to the BioChem. The internal cases were used to evaluate the searching ability of the planning algorithms while the external cases were used to evaluate the generalization capability of the model. The planning model is not a supervised learning model and does not learn how the true pathway is, so there is no leakage problem. It is worth noting that rule-based methods, such as RetropathRL, utilize all reaction templates and do not exclude those of test pathways for prediction. Our results on the internal and external sets and the added LASER set show that our model has good search and generalization capabilities. We reorganize the these parts by dividing the multi-step planning to internal testing and external testing in Results section.
- For the interpretation of Table 2, we used success rate and average solution to evaluate the quality of the output results, which is the basis of a multi-step bio-retrosynthesis planning model. And the hit rate of building blocks and pathway suggested the ability to output the reasonable and right results, respectively. The longest length represented that the model could deal with a complex structure with multi-step biosynthetic pathways. And finally the

consuming computational time was used to evaluate the computational cost. The ablation experiment summarized in Table 2 demonstrated that the current model architecture were optimal for multi-step bio-retrosynthesis planning. We have added these descriptions in Results section.

- L139: No information about why the RetroPathRL model was selected and whether it is applicable to the test data

➤ RetroPathRL is the most recent developed rule-based model for retro-biosynthesis, its reaction rules were extracted from all public biochemical reactions. As much as we known, this is the most close tool to BioNavi-NP, thus it is selected for comparison.

- L166: It is not the goal of the model to aim at developing something

➤ We have removed this statement there and discussed it later in Discussion section.

- L170: The authors test their models on 25 unseen NPs and find good performance. I'm sorry but I don't see any information on whether these compounds are easy or challenging to predict. I feel like being presented the unquestioned output of a black box

➤ Here the 25 external cases covered the different categories of natural products such as polyketides (case 18 and 23), terpenoids (case 14 and 17) and others. And the biosynthetic pathways of some cases have not been completely elucidated, due to the complex structures. BioNavi-NP have predicted the potential biosynthetic pathway for 22 of them and 17 of them hit the "right" building blocks according the prior knowledge, such as cases shown in Supplementary Figure 5. And we also test RetroPathRL with these cases using the default settings, only 2 of them retained with results. These suggested that our model shows better performance and greater generalization potential of our model than rule-based models.

- L173: here and elsewhere: please comment on the chance probabilities

➤ We revised the related sentences.

- L181: Again, at least I don't have a clue why this particular model/dataset was used at this point.

➤ In this work 40 structures that account for the origins of most NPs were selected as the building blocks (core library), and in some cases no pathways from the building blocks to target structures can be found by our model. Actually, many compounds in the early stage of the biosynthetic pathways including the well-known precursor metabolites (such as glycerate, adenine and so on) can be also treated as the building blocks since the routes from core library to these metabolites can be clear or these metabolites lie in the cytosol compartment of some microbial strain. The biosynthetic pathways from these metabolites to the target structure are also instructive, so the extended library was also supplied as an option for the users. We have fully revised the paragraph.

- L191: numbers are provided without any information/units

➤ This indicated the the number of candidate pathways the model output. It means that our model tend to output more alternatives than RetroPathRL. We corrected the sentence to "*BioNavi-NP outputs an average of 4.9 pathways, significantly higher than the 2.8 by*

RetroPathRL.”

- Caption of figure 2 is not well readable (zig-zag flow)

- More description has been added to the caption.

- L216: what's a CASP tool?

- CASP means computer-aided synthesis planning and used in the original literature, while this may cause ambiguity so we replace it with “*organic synthesis planning*” in the revised manuscript.

- L221: why have these compounds been selected for case studies?! What is their relationship to the training data. The test could be absolutely trivial

- The two cases are just selected to illustrate the usage of BioNav-NP. The first case (Sterhirsutin J) was not included in the training set and BioNavi-NP predicted the reasonable biosynthetic pathways. The second one was included in the training set, and BioNavi-NP successfully reproduced the known biosynthetic pathways and found a new potential pathway, which have been experimentally verified by a recent work.

- L251: the names of the options mentioned here are not consistent with the names shown in figure 3

- This have been fixed.

- L259: "In contrast to earlier works" could the authors please enlighten the reader what earlier works they are referring to? Is this statement referring to the most advanced “earlier works”, because there may always be “bad” earlier works?

- It means the rule-based retro-biosynthesis methods. We have corrected the statement in the revised manuscript.

- L261: "biased" seems to be used in a negative way, but bias may actually be positive in cases where the data are sparse

- We agree to this point and removed the statement.

- L307: "BioNavi-NP outperforms other computational tools" Many of the tools referred to in this work are known to not be applicable to biosynthesis prediction hence the statement likely is misleading

- We have removed the statement to “*BioNavi-NP shows a high success rate at generating biosynthetic routes and searching building blocks for NPs.*”

- L322: "The source code is available from the corresponding author upon reasonable request" This could mean anything. The authors should provide the source on defined terms via a public code repository

- The source code will be publicly available as our paper is accepted for publication, as stated in “Data and code availability” section. To protect our source code during the peer review procedure, it is not yet deposited in but will be uploaded to

<https://github.com/prokia/BioNavi-NP>. We have provided the source code for review process: https://drive.google.com/file/d/1wjGUG5SfPYi2LNa_dkOazHIPfO4YBLig/view. And the sentence claimed for code use has been revised.

- L337: Please explain what this company is doing otherwise the information on the conflict of interest is not interpretable

➤ The company provided the computational resources, and helped the engineering optimization of the retro* as well as the server development. Thus now we clearly added it to acknowledgement and contributions.

- Table 1: inconsistent use of N and n. And why are there no absolute numbers presented for Table 1 (they are reported in Table 2)?

➤ This has been fixed. We removed the absolute numbers in Table 2.

- L252-253, “Herein the outputs are redrawn to be clear (the raw output provided in Supplementary Figure 6-S30)”: which are the corresponding figures for the two examples in Figure 3b?

➤ The raw output for the first (Sterhirsutin J) and second case were provided as Supplementary Figure 26 and 32, respectively. This have been added to the revised manuscript.

- L270: I don’t understand why the Authors spend so many words on possible future directions while the manuscript falls short on many essential pieces of information to understand the current model

➤ Although we used “In future,” in that sentence, but the most content in the discussion section, is aiming to review our model and investigate its potent applications and more importantly, the shortcomings, to providing a better usage of it and guiding for how to improve the predicted result by combing other available tools. Now we have revised the manuscript carefully for those paragraph.

- L275-276: “...missing intermediates/building blocks if considering the fifth and sixth candidates separately”: the sixth candidate pathway is not marked and it is hard to understand what the authors are trying to say

➤ The mistake has been corrected. Here we tried to point out that each pathway of the model output should have been independent and complete, even for some complex structures originated from multiple building blocks, there should be multiple branches in the “right” pathway. While actually a specific pathway of the model output may miss some branches (as the case “Sterhirsutin J” discussed in the text), and the missing parts may just existed in other pathways, so we recommended users to consider all outputs and integrate them if necessary.

- L290-292: “When BioNavi-NP finds an unreasonable pathway in the first prediction, it is worth a second prediction to using a well-validated or user-confident intermediate as the target molecule.” Does “the first prediction” mean the first candidate pathway?

➤ It means the output with specific target structure as input. If the whole pathway looks unreasonable or incomplete based on prior knowledge, users are suggested to select a

reasonable intermediate from the output and submit another job with the intermediate as input.

- Are there duplicates between USPTO_NPL and BioChem? The description of “ the NP-like dataset containing 60000 reactions was organized from USPTO12 based on fingerprint similarity (threshold was 0.8) between components of reaction and NPs from the DNP database” in the supporting information is not clear what fingerprint was used and what exactly was kept. What about the similarity of 1.0?

➤ Yes, only 10 reactions were existed in both USPTO_NPL and BioChem. We add more description and revised the sentences as follow:

“Additionally, in order to learn more syntactic rules for chemical reactions, we screened reactions with components similar to natural products from the organic chemistry dataset for model pre-training. Specifically, we used molecular ECFP4 fingerprints to select the reactions with a maximum similarity of ≥ 0.8 between any component (reactants and/or products) of each reaction in the USPTO database and the NPs from the DNP database (version 27.2), resulting in 60,000 chemical reaction equations.”

Reviewer #3 (Expertise: bioengineering, metabolic engineering, syn bio):

Title: BioNavi-NP: Biosynthesis Navigator for Natural Products

Authors: Shuangjia Zheng, Tao Zeng, Chengtao Li, Binghong Chen, Connor W. Coley, Yuedong Yang, Ruibo Wu

Recommendation: Accept with minor revisions

Summary: An ensemble of four transformer neural networks supplemented by a multi-step Retro* algorithm is proposed as a method of hypothesizing biosynthesis pathways for natural products (NPs) for which these pathways are currently unknown. The four transformer neural networks predict a single biosynthesis step, hypothesizing the building blocks for the NP in question, with the Retro* strategy filling in the intermediate step details. The effects of training data and consideration of chirality is investigated. A few case studies are conducted. This work is linked to a public website which can be used by researchers to investigate potential biosynthesis pathways for organisms of interest.

General reviewer comments: Overall, this work presents great opportunities for the production of NPs, as NPs cannot be biomanufactured reliably without understanding their synthesis pathways. It appears the neural networks are more successful at predicting the building blocks of the NP than the Retro* strategy is at predicting the intermediate steps.

➤ We thank the reviewer for the great summary.

Major Concerns:

1) Overall: Biosynthesis of NPs, particularly those with peculiar products, are often restricted to a specific species, genus, or other biological subgroup. Synthesis pathways, even for the same or similar compounds, can vary greatly between dissimilar organisms, yet there is no discussion of the considerations of species or phylogeny of an organism when predicting biosynthesis pathways. The lack of consideration for the species in question for biosynthesis is, I believe, the greatest weakness of this work as presented. While adding the would be beyond the scope of the current work, a discussion of considering this as a future research direction would be appropriate.

- Thanks for very constructive suggestions, we have integrated an external tool, Selenzyme, into our service to take consideration of the enzyme and species problem. Please see details in the above summary response.

Minor Concerns:

1)TOC graphic: While the reference to Legos as “building blocks” of NP is clear, though most of the rest of your graphic is unclear. For instance, the transparency of the Lego blocks does not seem to have a purpose, neither does the compass underneath the desired NP. Further, it is unclear why the arrows move from the natural product to the basic building blocks, if it is desired to elucidate the NP biosynthesis pathway (arrows going in the other direction might make more sense). Overall, I think more words or labels are necessary in this figure to accurately convey meaning.

- We have revised the TOC. Since this work is focused on retro-biosynthetic routines prediction, the arrows direction are from target-NP to chemo-bricks (building blocks).

2)Discussion: While implementing this suggestion is beyond the scope of the current study, combining this tool with the prediction of the enzyme which could catalyze the predicted steps could be quite exciting and useful, as suggested in the discussion. This could also provide species- or genus- specific checks on the biological feasibility of pathways. Pathway likelihood could then be modulated by the enzymatic catalytic capabilities, as suggested in the manuscript’s discussion. If this could be added onto this BioNavi-NP tool, this could be a transformative and widely used tool. Would it be possible to incorporate model tools like or pfam or machine learning-based tools such as AlphaFold as a way to predict whether or not biocatalysis of a predicted reaction step is feasible? If this is feasible, consider including this in the discussion of enzymes in the discussion section, as this section is currently missing a discussion of the genome of the organism synthesizing the NPs in question, rather this tool just considers a general biosynthetic pathway.

- Very constructive suggestion! As described in summary response, Selenzyme has been integrated into our server to search enzymes for specific reaction. We did not use AlphaFold or pFam because they are not aiming to enzymatic function prediction.

3)Machine learning algorithm used. While in the manuscript considerations like chirality and training data for effectiveness of the BioNavi-NP tool, no discussion has been given to the machine learning algorithm used. Is a transformer neural network a standard for this type of application? Why was this particular method chosen over other machine learning techniques? How was the number of inputs or layers for these networks chosen? Further, it is not explained why an ensemble of four neural networks is used. Is this related to the four major classes of NPs discussed in the introduction?

- Motivation of the use of Transformer?

A: The Transformer neural network is one of the most popular models in the machine learning community today. It has not only shown SOTA capabilities in translation problems in the field of natural language processing, but recently has also shown powerful results in the problem of organic chemical synthesis. However, there are no works applied them into retro-biosynthesis tasks currently. Unlike the traditional template-based approach, this so-called template-free methods

bypass templates by learning a direct mapping from the SMILES representations of the product to reactants, providing greater generalization potential to complex compounds. This potential is well suited for biosynthesis, as the reactions are more complex and the mechanisms involved are not well understood. We have added this discussion in the revised Introduction section.

➤ How was the number of inputs or layers for these networks chosen; it's not explained why an ensemble of four neural networks is used.

A: The hyperparameter details of the model has been added in SI. The ensemble strategy of neural networks is a somehow regular trick for performance improvement of Transformer neural network. By integrating multiple models with different training steps, we can reduce the variance of the models and enable it to predict the results more accurately. This is not related to the classes of NP reactions. We have added more description in the caption of Figure 1.

Concerns of grammar, clarity, spelling, and similar:

1)Abstract. Throughout the abstract, errors in grammar and clarity are present, sometimes presenting the opposite meaning from what I believe is intended. Consider this:

a.Abstract. "More than 300,000 natural products (NPs) are discovered while fewer than 30,000 validated NPs compounds involved in about 33,000 known enzyme catalytic reactions are validated, and the complete biosynthesis pathways are uncovered for most NPs".

i.The first phrase ("More than 300,000 natural products (NPs) are discovered while fewer than 30,000 validated Ps compounds involved in about 33,000 known enzyme catalytic reactions are validated") is awkward to read. It may be best reformulated as "More than 300,000 natural products (NPs) have been discovered while fewer than 30,000 validated NP compounds involved in about 33,000 known enzyme-catalyzed reactions are validated".

ii.The second phrase ("and the complete biosynthesis pathways are uncovered for most NPs") means that the complete biosynthesis pathways are known for most NPs rather than, as you intend, to state that they are unknown. This would make your tool unnecessary. This could be phrased as "and the complete biosynthesis pathways are still unknown for most NPs."

b.This is just one example of unclear wording, sentence structure, or grammar. It is highly recommend this abstract is thoroughly proofread and edited. On the contrary, the body of the paper appears well-written and clear, so perhaps the writer of the body can proofread/edit the abstract.

➤ Thank you for point this out. We have re-organized the abstract to make it more clearer.

2)Page 7, line 174: plurality issue, original: "...and at least one building blocks were correctly identified...", corrected: "...and at least one building block was correctly identified...".

➤ We have revised the sentence accordingly.

3)Discussion section: The paragraphs stating the opportunities for improvement for BioNavi-NP should be labeled "(a)", "(b)", and so forth. Rather the last sentence at line 269 on page 11 should

be reformatted as an introductory sentence for the next few paragraphs, then the paragraphs do not need labels. Then each first sentence of these improvement paragraphs should be reworded as introductory sentences for the paragraphs.

➤ We have revised the sentences and formation.

4)Page 12 line 297: Original: “catalysis make it challenging to enzyme assignment”, correction: “catalysis make it challenging to assign enzymes” or similar.

➤ We have revised the sentence accordingly.

5) Please have another round of proofreading and editing for the paper to catch any issues I have not explicitly identified here.

➤ We have polished the whole manuscript as suggested.

References:

1. Moretti S, *et al.* Metanetx/mnxref: Unified namespace for metabolites and biochemical reactions in the context of metabolic models. *Nucleic Acids Res.* **49**, D570-D574 (2021).
2. Kim S, *et al.* Pubchem in 2021: New data content and improved web interfaces. *Nucleic Acids Res.* **49**, D1388-D1395 (2021).
3. Carbonell P, *et al.* Selenzyme: Enzyme selection tool for pathway design. *Bioinformatics* **34**, 2153-2154 (2018).
4. Liu B, *et al.* Retrosynthetic reaction prediction using neural sequence-to-sequence models. *ACS Cent. Sci.* **3**, 1103-1113 (2017).
5. Litsa EE, Das P, Kaviraki LE. Prediction of drug metabolites using neural machine translation. *Chem. Sci.* **11**, 12777-12788 (2020).
6. Pesciullesi G, Schwaller P, Laino T, Reymond JL. Transfer learning enables the molecular transformer to predict regio- and stereoselective reactions on carbohydrates. *Nat. Commun.* **11**, 4874 (2020).
7. Lin K, Xu Y, Pei J, Lai L. Automatic retrosynthetic route planning using template-free models. *Chem. Sci.* **11**, 3355-3364 (2020).
8. Rdkit: Open-source cheminformatics. <http://www.rdkit.org> (accessed 2018 Nov 29).

Reviewers' Comments:

Reviewer #1:

Remarks to the Author:

Major:

When I tried the platform, it did not produce any result for me (for Norcoclaurine, Oc1ccc(CC2NCCc3cc(O)c(O)cc32)cc1, the default parameters, Safari browser, Google chrome browser). I waited around 10 minutes for the results after the page was saying that the results were ready ("Job has finished, network is generating."), and the page kept reloading every several seconds, so it was not clear if the results were still being generated or the page was not working correctly. I repeated at least three more times without success. If it takes time for the results to be produced, please indicate on the webpage that the algorithm is in progress and give a link to download results when they are ready. If there are no pathways found, please, indicate this also immediately on the webpage.

Another search yielded two potential pathways for caffeine biosynthesis (default settings). Both pathways suggest formaldehyde as a direct precursor for caffeine, and other proposed intermediates are styrene-maleic anhydride and vinyl acetate. In terms of carbon conservation, the first pathway goes from C3 (pyruvate) to C12 to C11 to C1 to C8. Caffeine also has four nitrogen atoms, which appear out of nowhere. None of the proposed pathways make sense in the biochemist's eye. A quick check of the Selenzyme results for the last reaction of the proposed biosynthesis, formaldehyde to caffeine, shows that the most similar reaction is the unidirectional demethylation reaction of 3-methylxanthine to xanthine in the presence of NADPH + H⁺, yielding formaldehyde as a byproduct (<https://www.rhea-db.org/rhea/36291>). In my opinion, this example illustrates well the issue of BioNavi-NP to predict meaningful pathways: The model learns the two biotransformation 3-methylxanthine-xanthine and 3-methylxanthine-formaldehyde. As formaldehyde was not labeled as a cofactor, the second biotransformation is included in the training set, although formaldehyde should be labeled as a cofactor in this specific case. As the same happens for other demethylation reactions, the model learns that formaldehyde is connected to many other natural products, which is true but not very useful.

In most cases, formaldehyde is just a byproduct. Hence, the resulting pathways tend to take shortcuts through small cofactor-type molecules or byproducts, particularly those with an ambiguous role in biochemical pathways such as formaldehyde. The ambiguous roles of small molecules sometimes acting as substrates and sometimes as cofactors are well-known in the retrobiosynthesis community. It is disappointing that this issue is not addressed properly in BioNaviNP. While this may be out of the scope for this paper, I would still suggest looking into atom-mapping approaches to define correct main biotransformations in the training set.

I also tried to add glucose-6-phosphate as a precursor for predicting pathways toward caffeine, but I could not figure out where to specify building blocks for the prediction.

Finally, I would like to point out that some of the MetaNetX reaction references proposed by Selenzyme are deprecated, e.g., https://www.metanetx.org/cgi-bin/mnxweb/equa_info?equa=MNXR91754 or https://www.metanetx.org/cgi-bin/mnxweb/equa_info?equa=MNXR640.

p.4 Authors introduce the Selenzyme as the tool of choice for their platform. It would be nice if they could elaborate on why this tool was selected compared to other available tools for enzyme prediction for novel reactions.

p.7 The following sentence is not clear; please, elaborate.

"It should be noted that RetroPathRL had already contained all the reaction rules in MetaNetX so it also contained the rules in internal test cases"

In general, MetaNetX does not contain rules, RetroRules were automatically extracted from reactions

of MetaNetX by authors of RetroPath. What do you mean "it contained the rules in internal test cases"?

p.8 The authors say that the longest pathway their algorithm can predict is six reaction steps. Can authors elaborate on how they advise using the algorithm for predicting the biosynthesis routes for the molecules that require longer pathways in the Discussion section? Is the six steps fundamental limitation of the NN models selected, or will it be possible to predict longer pathways with the same model and better computational power?

In addition, I believe the manuscript will benefit from adding the table explicitly comparing the method of pathway prediction proposed by authors with other state-of-the-art methods available to the community for pathway prediction, at least with the 5-7 most cited methods other than RetroRathRL. For example, authors can compare the ways for predicting the reaction steps, availability (online, code), longest possible pathway, range of molecules, ways for ranking., etc., for these methods.

Minor:

p.3 I would recommend putting a citation for this statement:
"NPs are often the best option for seeking novel bioactive templates in human health as many of them are critical factors for regulating intrinsic biofunctions, especially in plants. "

p.3 "organizing pathways by similarity matching with known reactions":
It is not clear to me what is meant here.

p.4 Authors talk about an increase in accuracy related to integrating the organic chemistry reactions to their model (from 10.6% to 17.2% for top-1 and from 27.8% to 48.2% for top-10). I think it is misleading to claim "the >60% increase in accuracy", here the accuracy increased by 6.6% and 20.4% correspondingly. Please, reformulate the sentence so that the percentages of the total / of the previous results are not mixed.

p.3. "a bunch of rules"
Suggestion: replace bunch with collection

p.4 "While these methods are more flexible, they still have difficulty in predicting unknown reaction types."
As the authors do not address the prediction of unknown reaction types, this statement should be revisited.

p.4 "several intermediates are underwent."
This is not clear.

Issues remain with style and spelling to the point that it is difficult for the reader to extract the message. I would highly recommend another round of professional editing and grammar check. Many commas are missing in the text.

Examples of the remaining spelling / style issues (there might be more in the text):
Abstract, p.2 : a iterative
p.4 the predicted reaction cannot happen even it obeys the reaction rule (if missing)
p.4 free models has shown
p.4 search technicque
p.4 netwrok
p.6 a optimal

p.7 Part "Multi-step planning: Internal testing" is written in present tense. Please, rewrite in the past tense.

p.14 "for the rest reactions, we derived pairs of precursors and metabolites"
Did you mean rest of the reactions?

Reviewer #2:

Remarks to the Author:

I appreciate the efforts made by the Authors to address my comments and improve the quality of this manuscript. However, it seems that a few issues have not been fully addressed yet.

--

(1) Language issues remain a major concern. Many parts are difficult to read (in particular the newly added parts); some are unclear or even misleading. Professional language editing will be required.

Example: Abstract

- "fewer than 30,000 of them are reported to be involved in about 33,000 known enzyme-catalyzed reactions" Statement is unclear and should read "involved in a total of about..." ...otherwise one could interpret this in the way that 33,000 NPs are involved in ANY of the 33,000 known enzyme-catalyzed reactions.
- Long sentences that are difficult to read and feel out of place (they are disconnected from the preceding and succeeding sentences): "First the single-step bio-retrosynthesis prediction was made by an enhanced molecular Transformer neural network, which is trained by combining general organic and biosynthetic reactions, then the plausible biosynthetic pathways can be efficiently sampled with an AND-OR tree-based planning algorithm from a iterative multi-step bio-retrosynthetic routes."
...and further down the road:
- "this vast chemical space of NPs is reachable from dozens of simple building blocks..." I believe that what the Authors actually mean to say is that "this vast chemical space of NPs is spanned by just a few dozen, simple building blocks"
- "To date, many efforts have been made in the field of retro-biosynthesis^{11, 12} through organizing pathways by similarity matching with known reactions or predicting pathways by rule-based model. The former methods...": First sentence is very difficult to read and the latter part of the first sentence is not good English. It is also not clear what "former methods" refers to exactly.
- "These methods are limited in their applications to complex NPs by the scope of the database". I believe that what the Authors actually mean to say is something like "In particular for complex NPs, these methods are limited in their scope...".
- "While these methods are more flexible".... more flexible than what?
- "our model solved 94.7%... of the cases". It is not clear what "solved" (and "cases" mean (avoid these terms).
- "Additionally, we found that correct handling of stereochemical information is essential for biosynthesis, as removal of chirality from the reaction SMILES decreases the top-10 accuracy from 27.8% to 16.3% (BioChem (w/o chirality))." It is difficult to understand where the part "(BioChem (w/o chirality))" belongs to.
- ...and the SI:
- Table 1 "Compare of..." error. Also, it should be SI Table, not just Table.
- "Supplementary Figure 1. The building blocks used in BioNavi-NP." Building blocks used for what?

The manuscript is also ripe with grammatical errors and typos. Use of tense is inconsistent throughout the manuscript. Sentences starting with "And" should be avoided.

It is not possible for me to list all language issues.

--

(2) There are still many instances where numbers/statements are not accompanied by sufficient information that would allow their interpretation.

Example: Abstract

- "pathways for 90% of test compounds and recovering the verified building blocks for 73%, significantly outperforming conventional rule-based approaches": what if the number of test compounds were just 10?! Please add information and also, please use the term "significant" only in the context of statistical significance.

- "Moreover, BioNavi-NP also shows an outstanding enumeration capacity of biologically-plausible pathways.": There's not really any information provided with this statement because the Authors miss to say anything about what "outstanding enumeration capacity" means (in terms of numbers) and how exactly this was determined.

Example Results:

- Table 1: It is unclear during which type of experiment these numbers were obtained (test set? cross-validation?) It is still unclear how accuracy is defined in the specific context. Table 1 is not really interpretable, and the corresponding text in the main text isn't either: "As summarized in Table 1, the Transformer models directly trained on the BioChem dataset of 31710 reactions achieves top-1 and 10 accuracies of 10.6% and 27.8%, respectively"

(3) TOC graphics: I'm sorry but to me this figure is still not informative and it is difficult to interpret. I mean, what message are the parts that are faded out supposed to convey (and this is just one example of a question that comes up when I look at this TOC graphic)?

(4) One of my previously mentioned points is that the authors should "please comment on the chance probabilities". What I meant to ask is, could the authors please report chance probabilities to have a baseline that give a better understanding of the value/performance of the models (in other words, the difficulty of the test cases)?

(5) Can the Authors please investigate and discuss how the model performs on NPs from different kingdoms?

(6) Another open issue seems to be the definition of the applicability domain of the model. How is it defined and what measures are in place to avoid the presentation of unreliable predictions to users of the web service (this should be part of the manuscript)?

Minor: Figure legend of SI Figure 4: Should read "Internal cases"

Reviewer #3:

Remarks to the Author:

General reviewer comments: As before, I think this tool shows great promise for the development of synthetic pipelines for NP synthesis and it holds great potential through the identification of system pathways. The inclusion of selenzyme to screen for enzyme feasibility is a good method for addressing previous reviewer comments.

Major Concerns:

1) The reliance of BioNavi-NP on Retro* and its ensemble. In the introduction, it is stated that "the Retro* strategy is used to select the top branches reaction for each intermediate", whereas BioNavi-NP is stated as having an input of the NP and an output of the precursors. Also, somewhere, there is an ensemble layer, which figure 1 seems to indicate that only a single branch of the ensemble was used to create BioNavi-NP. So how dependent is BioNavi-NP on Retro* and the ensemble step? The paragraph in lines 108 to 120 poorly describes this process and Figure 1B does not seem to help much. Overall, the relationship of BioNavi-NP to previous algorithms is unclear. Further, was the

ensemble layer created in this work or another work? Retro* clearly comes from a previous work, and BioNavi-NP from this work, but this middle step is unclear.

2) Lines 230-239: Given the abundance of metabolic products created from amino acids, it is surprising that amino acids have the lowest correct pathway and building block scores. How does the performance metrics in Figure 2b compare to the abundance of training data (e.g. what fraction of the training data corresponded with each category of NP). Is this truly a lack of data problem (suggested in line 239), an issue of difficult decomposition (suggested in lines 235 to 238), or attributable to both?

3) Assessment of the tool: In some places, the assessment criteria for the tool seem to be low bars that would be still impractical to implement in vivo or requires a lot of test runs in vitro. For instance, in lines 241 to 250, in 68% of cases, at least one building block was correctly identified. This seems like a low threshold considering that three or more building blocks are needed for many of the case studies. What are the rates for all building blocks being correct? If this rate is low, why would it be advantageous to have a system like BioNavi-NP which might only identify one correct precursors across five or more different synthesis pathways, but will still need experimental validation? I think this issue needs to be addressed in the manuscript.

4) Discussion. While a frank discussion of the limitation of the current tool is appreciated, this discussion section appears to lack much of a review of the improvements that BioNavi-NP brings to this area of research and seems to focus too much on the limitations. Consider the phrase on lines 346 to 347: "BioNavi-NP seems still inadequate", which seems to beg the question, if the tool is inadequate, why publish it now and in such a prestigious journal? As I have said before, I believe the work holds great potential, but the lack of credit given to the tool in the discussion section seems to undercut its inclusion in such a prestigious journal as Nature Communications. Suggest rewriting.

5) Energy costs. In many organisms, the energy costs associated with NP production either limit its production (e.g. not enough energy to produce something) or drive its production (e.g. the production of the NP serves as an electron or energy dump). Does BioNavi-NP consider or predict cofactors necessary for each predicted biosynthesis step?

Minor Concerns:

1) Lines 218-221: This reads that for a set of desired natural products, BioNavi-NP on average produces more alternative pathways for synthesis than does RetroPathRL (up to a maximum of 5). Then it says that BioNavi-NP has a longer pathway than does RetroPathRL in its predictions. The claim is then made that BioNavi-NP is able to more effectively address complex NPs. This last claim does not appear to be strictly true for this particular comparison. This shows only that BioNavi-NP can produce more hypothetical synthesis pathways with more complexity than RetroPathRL, not that it is more efficient. A "more efficient" claim would need to be justified with computing time required to get the results. Also unclear here is how many NPs were used for this analysis.

2) Web interface. Does the web interface allow selection of precursor metabolites, or are these pre-defined?

3) Tool. Does the tool consider non-enzymatic steps?

Concerns of grammar, clarity, spelling, and similar:

1) Citations. The in-text citation style is somewhat inconsistent, in that in line 65 the citation occurs after the period, yet in line 66 the citation occurs before the period. Please make consistent.

2) Proofreading. There are still a few areas in need of proofreading, for example (line 69): "commercially-relevant quantities of NPs is a major obstacle to their therapeutic translation." Should be "commercially-relevant quantities of NPs are a major obstacle to their therapeutic translation." Another example is line 92: "predicted reaction cannot happen even it obeys the reaction rule", should be "predicted reaction cannot happen even if obeys the reaction rule". Spelling mistakes also are still present, such as in line 119 "network". Formatting mistakes are also present such as in line 338 where there is a period on either side of a reference. Please make sure to proofread your article again.

Reviewer #1 (Remarks to the Author):

Major:

When I tried the platform, it did not produce any result for me (for Norcoclaurine, Oc1ccc(CC2NCCc3cc(O)c(O)cc32)cc1, the default parameters, Safari browser, Google chrome browser). I waited around 10 minutes for the results after the page was saying that the results were ready ("Job has finished, network is generating."), and the page kept reloading every several seconds, so it was not clear if the results were still being generated or the page was not working correctly. I repeated at least three more times without success. If it takes time for the results to be produced, please indicate on the webpage that the algorithm is in progress and give a link to download results when they are ready. If there are no pathways found, please, indicate this also immediately on the webpage.

- Thanks for the reviewer's comments. Through testing, we found that this bug occasionally appears on the Mac system. In fact, the results you submitted were successfully predicted in the backend, but just did not show up. We have placed the results below (or you can check it in <http://biopathnavi.qmclab.com/job.html?JobId=4L1MYGAYEHG3X5TLBDG1>). To address the questions raised by the reviewer, we have modified the webpage by adding a prompt that indicates the progress and allows users to retrieve the results manually.

Another search yielded two potential pathways for caffeine biosynthesis (default settings). Both pathways suggest formaldehyde as a direct precursor for caffeine, and other proposed intermediates are styrene-maleic anhydride and vinyl acetate. In terms of carbon conservation, the first pathway goes from C3 (pyruvate) to C12 to C11 to C1 to C8. Caffeine also has four nitrogen atoms, which appear out of nowhere. None of the proposed pathways make sense in the biochemist's eye. A quick check of the Selenzyme results for the last reaction of the proposed biosynthesis, formaldehyde to caffeine, shows that the most similar reaction is the unidirectional demethylation reaction of 3-methylxanthine to xanthine in the presence of NADPH + H⁺, yielding formaldehyde as a byproduct (<https://www.rhea-db.org/rhea/36291>). In my opinion, this example illustrates well the issue of BioNavi-NP to predict meaningful pathways: The model learns the two biotransformation 3-methylxanthine-xanthine and 3-methylxanthine-formaldehyde. As formaldehyde was not labeled as a cofactor, the second biotransformation is included in the

training set, although formaldehyde should be labeled as a cofactor in this specific case. As the same happens for other demethylation reactions, the model learns that formaldehyde is connected to many other natural products, which is true but not very useful. In most cases, formaldehyde is just a byproduct. Hence, the resulting pathways tend to take shortcuts through small cofactor-type molecules or byproducts, particularly those with an ambiguous role in biochemical pathways such as formaldehyde. The ambiguous roles of small molecules sometimes acting as substrates and sometimes as cofactors are well-known in the retrobiosynthesis community. It is disappointing that this issue is not addressed properly in BioNaviNP. While this may be out of the scope for this paper, I would still suggest looking into atom-mapping approaches to define correct main biotransformations in the training set.

- Thanks for pointing this out. This problem is mainly because the training dataset has inherent defects. Many specialized metabolic reactions specified in MetaCyc pathway ontology and KEGG pathway map are not mass-balanced, or contain generic compounds (like co-factors) that prevent accurate simulations within models. We noticed this problem and have made efforts in data pre-processing to reduce the presence of cofactors (detailed in Methods section), allowing 89% of the training pairs to be singletons (i.e. only one reactant and one product).

We agree with the reviewer that the mass-balance is important to further improve the quality of this work. However, in this work we focused on highlighting the potentials of rule-free method and had shown better performance than rule-based bio-retrosynthesis baselines statistically. The combination of atom mapping approaches with deep learning methods is definitely a promising future direction for biosynthesis but is beyond the scope of this work. Its application in the field of biosynthesis is still unexplored and will be an interesting topic. We leave it for future research. We added this in Discussion as:

“Furthermore, a few predicted pathways tend to take shortcuts through small cofactor-type molecules or by-products. In fact, efforts have been made in data pre-processing to reduce the ambiguous role of generic compounds, but there are still a small fraction of anomalous data leading to unjustified shortcuts. Future work to combine atom-mapping approaches will allow better mass-balance for the single-step prediction.”

I also tried to add glucose-6-phosphate as a precursor for predicting pathways toward caffeine, but I could not figure out where to specify building blocks for the prediction.

- We have added a new function for specifying building blocks, while this may not work well if the new precursors are not (or rarely) seen in our training set. We have added this notice in the corresponding place of the web interface.

Finally, I would like to point out that some of the MetaNetX reaction references proposed by Selenzyme are deprecated, e.g., https://www.metanetx.org/cgi-bin/mnxweb/equa_info?equa=MNXR91754 or https://www.metanetx.org/cgi-bin/mnxweb/equa_info?equa=MNXR640.

- Thank you for pointing this out. The predicted reactions were directly obtained from the server of Selenzyme, and we have no privilege to change the reaction templates. We added one sentence in the method section.

“We noticed that Selenzyme occasionally outputs deprecated reaction references, and we kept it as well for reference.”

p.4 Authors introduce the Selenzyme as the tool of choice for their platform. It would be nice if they could elaborate on why this tool was selected compared to other available tools for enzyme prediction for novel reactions.

- Thanks for your valuable comments. We summarized the tools that can be easily called at present, and found that Selenzyme is the fastest and most convenient to call. We have added a description of this in the revised manuscript:

“For a predicted biosynthetic pathway, it is crucial to identify the enzymes that will enable the reaction to take place. Several tools exist to select candidate enzymes for novel reactions, including Selenzyme, BridgIT, and BRENDA. We introduced Selenzyme to search for candidate enzymes for specific reactions because of its lightweight RESTful service and additional biological metadata.”

p.7 The following sentence is not clear; please, elaborate.

"It should be noted that RetroPathRL had already contained all the reaction rules in MetaNetX so it also contained the rules in internal test cases"

In general, MetaNetX does not contain rules, RetroRules were automatically extracted from reactions of MetaNetX by authors of RetroPath. What do you mean "it contained the rules in internal test cases"?

- We want to show that the templates of RetroPathRL have considered all of the reactions in our test set, proving that our comparison is fair. We have revised this sentence to *“It should be noted that RetroPathRL contains the reaction rules extracted from MetaNetX, while all the internal test NPs were included in MetaNetX.”*

p.8 The authors say that the longest pathway their algorithm can predict is six reaction steps. Can authors elaborate on how they advise using the algorithm for predicting the biosynthesis routes for the molecules that require longer pathways in the Discussion section? Is the six steps fundamental limitation of the NN models selected, or will it be possible to predict longer pathways with the same model and better computational power?

- Six steps is not the maximum that the model can handle, but the longest pathway in our test set. For molecules requiring longer pathways than six steps, BioNav-NP is able to give predictions. For example, the longest pathway for case 18 and case 19 of external cases (see below, also depicted in Supplementary Figure 24 and 25, respectively) is 7 and 8 respectively. Theoretically there is no fundamental limitation for the length of output pathway, while generally the longer the path, the lower the ranking will be, and also, the more time consumption will be. 10 reaction steps is the default termination condition of our model and it can be set to a maximum of 20 by the option of “Max_depth”.

We clarified the sentence as: *“The length of the pathways is affected by the “Max_depth” (default is 10). One can increase the number to a maximum of 20 upon request.”*

Case 18

Case 19

In addition, I believe the manuscript will benefit from adding the table explicitly comparing the method of pathway prediction proposed by authors with other state-of-the-art methods available to the community for pathway prediction, at least with the 5-7 most cited methods other than RetroRathRL. For example, authors can compare the ways for predicting the reaction steps, availability (online, code), longest possible pathway, range of molecules, ways for ranking., etc., for these methods.

- Thank you for your suggestion, we have summarized related methods and listed them in the revised manuscript as a supplementary table (Supplementary Table 1), see below.

Name	Method	Planning strategy	Data source	Pathway ranking	Stereo-chemistry	Availability	Description
MRE	Reaction search	Enumeration	KEGG	Host-dependent score	✓	Webservice	Guiding the design and optimization of heterologous biosynthesis pathways
RouteSearch		Branch-and-Bound	MetaCyc	Reaction costs Atom mapping cost	✓	Software	Computing the optimal metabolic routes in genome-scale reaction networks
PathPred	Rule-based	Enumeration	KEGG	Structure similarity	✓	Webservice	Predicting enzyme-catalyzed metabolic pathway based on RDM patterns
Novostoic		Enumeration	MetRxn KEGG ...(multiple)	Gibbs free energy	✓	Code	Designing bioconversion routes while considering mass conservation, cofactor balance
Envipath		Enumeration	EAWAG-BBD	Aerobic likelihood	-	Webservice	Predicting the microbial biotransformation of organic environmental contaminants
BNICE.ch		Enumeration	KEGG	Gibbs free energy Pathway length	-	Webservice	Allowing the de novo synthesis of metabolic pathways with thermodynamic properties
RetroPathRL		MCTS	MetaNetX	Structure similarity Sequence diversity	Option	Code	Exploring the bioretrosynthesis space using reinforcement learning for metabolic engineering
Bionavi-np	Rule-free	Retro*	KEGG MetaCyc ...(multiple)	Reaction cost Host-dependent score ...(multiple)	✓	Code Webservice	Navigating retrobiosynthesis pathways for natural products with no need of reaction rules

Minor:

p.3 I would recommend putting a citation for this statement:

"NPs are often the best option for seeking novel bioactive templates in human health as many of them are critical factors for regulating intrinsic biofunctions, especially in plants. "

➤ Added accordingly.

p.3 "organizing pathways by similarity matching with known reactions":

It is not clear to me what is meant here.

➤ It means that some methods perform pathway search based on existing reaction databases according to chemical similarity. We now changed it as:

"To date, many efforts have been made in the field of retro-biosynthesis, which can be

roughly divided into two categories of methods: knowledge-based and rule-based approaches (Supplementary Table 1 provides an overview on popular methods). The knowledge-based approaches enumerate possible biosynthesis routes according to the existing reaction databases (such as MetaCyc and KEGG) and rank the suggested routes through scoring functions such as chemical similarity and chassis.”

p.4 Authors talk about an increase in accuracy related to integrating the organic chemistry reactions to their model (from 10.6% to 17.2% for top-1 and from 27.8% to 48.2% for top-10). I think it is misleading to claim "the >60% increase in accuracy", here the accuracy increased by 6.6% and 20.4% correspondingly. Please, reformulate the sentence so that the percentages of the total / of the previous results are not mixed.

- This have been revised to
“When 60K organic reactions involving natural product-like compounds were joined for training (BioChem + USPTO_NPL), the top-1 and 10 accuracies of the model raised to 17.2% and 48.2%, respectively. The large increase in accuracies by data augmentation indicated that organic reaction expertise was helpful for accurately predicting biological steps.”

p.3. "a bunch of rules"

Suggestion: replace bunch with collection

- Revised accordingly.

p.4 "While these methods are more flexible, they still have difficulty in predicting unknown reaction types."

As the authors do not address the prediction of unknown reaction types, this statement should be revisited.

- Thank you for pointing out this, we have revised the sentence as:
“Although rule-based methods have led to promising results in retro-biosynthesis, a few challenges remain. First, their ways to formulate expert-approved rules are complicated and time-consuming. Second, the degree of generality/specificity of curated rules may lead to invalid or incomplete proposals. Third, they are fundamentally unable to predict novel reactions beyond the rule databases.”

p.4 "several intermediates are underwent."

This is not clear.

- This has been removed.

Issues remain with style and spelling to the point that it is difficult for the reader to extract the message. I would highly recommend another round of professional editing and grammar check. Many commas are missing in the text.

Examples of the remaining spelling / style issues (there might be more in the text):

Abstract, p.2 : a iterative

p.4 the predicted reaction cannot happen even it obeys the reaction rule (if missing)

p.4 free models has shown

p.4 search technique

p.4 network

p.6 a optimal

p.7 Part "Multi-step planning: Internal testing" is written in present tense. Please, rewrite in the past tense.

- Thank you very much for the comment. We have corrected all that you suggested and done a new proofreading. The updated manuscript has further been polished by a native English-speaking expert in cheminformatics.

p.14 "for the rest reactions, we derived pairs of precursors and metabolites"

Did you mean rest of the reactions?

- Yes, it has been revised.

Reviewer #2 (Remarks to the Author):

I appreciate the efforts made by the Authors to address my comments and improve the quality of this manuscript. However, it seems that a few issues have not been fully addressed yet.

--

(1) Language issues remain a major concern. Many parts are difficult to read (in particular the newly added parts); some are unclear or even misleading. Professional language editing will be required.

Example: Abstract

- "fewer than 30,000 of them are reported to be involved in about 33,000 known enzyme-catalyzed reactions" Statement is unclear and should read "involved in a total of about..." ...otherwise one could interpret this in the way that 33,000 NPs are involved in ANY of the 33,000 known enzyme-catalyzed reactions.

- Revised accordingly.

- Long sentences that are difficult to read and feel out of place (they are disconnected from the preceding and succeeding sentences): "First the single-step bio-retrosynthesis prediction was made by an enhanced molecular Transformer neural network, which is trained by combining general organic and biosynthetic reactions, then the plausible biosynthetic pathways can be efficiently sampled with an AND-OR tree-based planning algorithm from a iterative multi-step bio-retrosynthetic routes."

- This have been revised to

"First, a single-step bio-retrosynthesis prediction model was trained using both general organic and biosynthetic reactions through end-to-end Transformer neural networks. Based on this model, plausible biosynthetic pathways can be efficiently sampled through an AND-OR tree-based planning algorithm from iterative multi-step bio-retrosynthetic routes."

...and further down the road:

- "this vast chemical space of NPs is reachable from dozens of simple building blocks..." I believe

that what the Authors actually mean to say is that “this vast chemical space of NPs is spanned by just a few dozen, simple building blocks”

➤ Yes, it has been changed to

“Remarkably, this vast chemical space of NPs is reachable from a few dozen, simple building blocks”

- “To date, many efforts have been made in the field of retro-biosynthesis^{11, 12} through organizing pathways by similarity matching with known reactions or predicting pathways by rule-based model. The former methods...”: First sentence is very difficult to read and the latter part of the first sentence is not good English. It is also not clear what “former methods” refers to exactly.

➤ This has been revised to

“To date, many efforts have been made in the field of retro-biosynthesis, which can be roughly divided into two categories of methods: knowledge-based and rule-based approaches (Supplementary Table 1 provides an overview on popular methods). The knowledge-based approaches enumerate possible biosynthesis routes according to the existing reaction databases (such as MetaCyc and KEGG) and rank the suggested routes through scoring functions such as chemical similarity and chassis.”

- “These methods are limited in their applications to complex NPs by the scope of the database”. I believe that what the Authors actually mean to say is something like “In particular for complex NPs, these methods are limited in their scope...”.

➤ We have revised it to

“For complex NPs, these methods are often not applicable when the reactions of their biosynthetic pathways are not included in those databases.”

- “While these methods are more flexible”.... more flexible than what?

➤ It means that rule-based methods are more flexible than the approaches that enumerate biosynthesis routes based on the existing reaction databases. The sentence has been changed to

“Although rule-based methods have led to promising results in retro-biosynthesis, a few challenges remain.”

- “our model solved 94.7%... of the cases”. It is not clear what “solved” (and “cases” mean (avoid these terms).

➤ The sentence has been changed to

“our model achieved a hit rate of 94.7% (144/152) on the LASER test set”

- “Additionally, we found that correct handling of stereochemical information is essential for biosynthesis, as removal of chirality from the reaction SMILES decreases the top-10 accuracy from 27.8% to 16.3% (BioChem (w/o chirality)).” It is difficult to understand where the part “(BioChem (w/o chirality))” belongs to.

➤ The sentence has been changed to

“The correct handling of stereochemical information contributed to the prediction of biosynthetic reaction, as the removal of chirality from the reaction SMILES decreased the top-10 accuracy from 27.8% (BioChem) to 16.3% (BioChem (w/o chirality)).”

...and the SI:

- Table 1 “Compare of...” error. Also, it should be SI Table, not just Table.

➤ Revised accordingly.

- “Supplementary Figure 1. The building blocks used in BioNavi-NP.” Building blocks used for what?

➤ Used as the end point of the pathway search. This caption has been revised to *“The building blocks of core library used as the end point of the pathway search.”*

The manuscript is also ripe with grammatical errors and typos. Use of tense is inconsistent throughout the manuscript. Sentences starting with “And” should be avoided. It is not possible for me to list all language issues.

➤ Thank you very much for the comment. We have corrected all that you suggested and done a new proofreading. The updated manuscript has further been polished by a native English-speaking expert in cheminformatics.

(2) There are still many instances where numbers/statements are not accompanied by sufficient information that would allow their interpretation.

Example: Abstract

- “pathways for 90% of test compounds and recovering the verified building blocks for 73%, significantly outperforming conventional rule-based approaches”: what if the number of test compounds were just 10?! Please add information and also, please use the term “significant” only in the context of statistical significance.

➤ We have removed the “significant” accordingly and the sentence has been changed to *“Extensive evaluations revealed that BioNavi-NP could identify biosynthetic pathways for 90.2% of 368 test compounds and recovered the building blocks for 72.8%, 1.7 times more accurate than the conventional rule-based approaches.”*

- “Moreover, BioNavi-NP also shows an outstanding enumeration capacity of biologically-plausible pathways.”: There’s not really any information provided with this statement because the Authors miss to say anything about what “outstanding enumeration capacity” means (in terms of numbers) and how exactly this was determined.

➤ This has been removed and revised to *“The model was further shown to identify biologically plausible, novel pathways for complex NPs collected from the recent literature.”*

Example Results:

- Table 1: It is unclear during which type of experiment these numbers were obtained (test set? cross-validation?) It is still unclear how accuracy is defined in the specific context. Table 1 is not

really interpretable, and the corresponding text in the main text isn't either: "As summarized in Table 1, the Transformer models directly trained on the BioChem dataset of 31710 reactions achieves top-1 and 10 accuracies of 10.6% and 27.8%, respectively"

- We have added more details of evaluation methods as:

"To train our model, we first curated a biosynthesis data set from the public database called BioChem (see Methods), containing 33710 unique pairs of precursors and metabolites. From the dataset, we randomly selected 1000 pairs as the test set, 1000 pairs as the validation set, and the remaining as the training set. Following the previous retrosynthesis works, we evaluated the performance on the test set using top-n accuracies, defined as the percentages of correct instances among top-n predicted precursors."

Also, we have rewritten the text of Table 1 to make it clearer.

(3) TOC graphics: I'm sorry but to me this figure is still not informative and it is difficult to interpret. I mean, what message are the parts that are faded out supposed to convey (and this is just one example of a question that comes up when I look at this TOC graphic)?

- We have revised the TOC graphics to make it clearer, as shown below.

(4) One of my previously mentioned points is that the authors should "please comment on the chance probabilities". What I meant to ask is, could the authors please report chance probabilities to have a baseline that give a better understanding of the value/performance of the models (in other words, the difficulty of the test cases)?

- Thanks for your valuable comments. From a machine learning perspective, the approach to avoiding chance ability is primarily by evaluating the model with internal and/or external test sets from different sources to see if there is consistent performance. Specifically, in this work, we have taken various types of test data including random split internal test set, external test sets reported by RetroPathRL (Laser and Golden set), and external cases from literature to evaluate our model. The types and sources of molecules among these datasets are diverse (see Supplementary Figure 4) and the results are consistently comparable or better than baselines. We believe we have minimized the potential for oversimplification of the test sets.

(5) Can the Authors please investigate and discuss how the model performs on NPs from different kingdoms.

- Good point. To perform this analysis, the kingdom of NPs in the internal test set (a total of 368 cases) were further annotated **manually** from MetaCyc and literature references according to their biosynthetic pathway, which can be divided into plant, animal, bacteria, fungi and others (such as some algae like Bacillariaceae). The kingdom distribution and model performance were shown in the table and figure below (also depicted in the SI as Supplementary Table 2 and Supplementary Figure 5, respectively). We observed that BioNavi-NP performed the best on the NPs from the plant, and the worst on the NPs from others and fungi, where the NPs from the AAs biosynthetic pathway take the majority. This has also been added to the manuscript as:

“The model performance on NPs from different kingdoms was further investigated (Supplementary Figure 5), where the NPs in the internal set were manually divided into plant, bacteria, fungi, animal and others (such as some algae like Bacillariaceae). The results showed that the model achieved the highest hit rate of pathway and building blocks (30.7% and 82.0%, respectively) for the plant kingdom. This is consistent with our results at the pathway level because the plant NPs mainly fall into the MVA/MEP and CA/SA categories (Supplementary Table 2). By contrast, NPs from the fungi and others had lower hit rate of building blocks (46.1% and 41.7%, respectively), mainly due to the fact that the majority of them in the internal set are NRPs derived from the AAs pathway.”

Supplementary Table 2. The distribution of internal test set in different kingdoms and pathways.

	Plant	Bacteria	Fungi	Animal	Others	Total
AA/MA	13	24	0	0	0	37
AAs	13	42	25	6	8	94
CA/SA	73	3	0	13	2	91
MVA/MEP	102	19	11	9	0	141
Others	4	6	3	0	2	15
Total	205	94	39	28	12	378 ^a

^a Some NPs are distributed in multiple kingdoms, and they were counted repeatedly, resulting in inconsistencies between the number of cases in the manuscript (368) and the total number here (378).

Supplementary Figure 5. The BioNavi-NP's performance within each kingdom.

(6) Another open issue seems to be the definition of the applicability domain of the model. How is it defined and what measures are in place to avoid the presentation of unreliable predictions to users of the web service (this should be part of the manuscript)?

➤ Theoretically any valid natural product structure can be used as an input to the BioNavi-NP. However, as shown in Figure 2, there are differences in the performance of the model for different natural products families due to the bias and coverage of the training data, which has been well discussed in the manuscript. In order to avoid unreliable outputs, we have made extra efforts in the following aspects:

i) Modules that link the predicted intermediates and reactions to the known data sources were added to the webserver to offer annotation and facilitate the re-ranking of pathways.

ii) The introduction of enzyme prediction tool, this will be a further evaluation for the biological feasibility of these pathways according to the estimated preference of species and enzymes.

iii) The result of BioNavi-NP web service is organized by an interactive tree network where all plausible pathways are integrated. Users can evaluate all candidate pathways from a holistic perspective and get the complete pathway by merging some of them.

These features have been addressed in the Discussion section to enable experts to better understand and filter the output of BioNavi-NP. Finally, we would like to stress that although BioNavi-NP is an effective tool for computer-aided biosynthesis, the experience and knowledge of experts still play an important role. Therefore, manual checks and corrections are necessary and highly recommended.

Minor: Figure legend of SI Figure 4: Should read "Internal cases"

➤ Revised accordingly.

Reviewer #3 (Remarks to the Author):

General reviewer comments: As before, I think this tool shows great promise for the development of synthetic pipelines for NP synthesis and it holds great potential through the identification of

system pathways. The inclusion of selenzyme to screen for enzyme feasibility is a good method for addressing previous reviewer comments.

➤ We appreciate the review's positive comments.

Major Concerns:

1) The reliance of BioNavi-NP on Retro* and its ensemble. In the introduction, it is stated that “the Retro* strategy is used to select the top branches reaction for each intermediate”, whereas BioNavi-NP is stated as having an input of the NP and an output of the precursors. Also, somewhere, there is an ensemble layer, which figure 1 seems to indicate that only a single branch of the ensemble was used to create BioNavi-NP. So how dependent is BioNavi-NP on Retro* and the ensemble step? The paragraph in lines 108 to 120 poorly describes this process and Figure 1B does not seem to help much. Overall, the relationship of BioNavi-NP to previous algorithms is unclear. Further, was the ensemble layer created in this work or another work? Retro* clearly comes from a previous work, and BioNavi-NP from this work, but this middle step is unclear.

➤ BioNavi-NP is the combination of Retro* and ensemble Transformer neural networks in the context of retro-biosynthesis. The Transformer neural network is used as a single-step prediction model to generate the candidate precursors for a target NP, while the ensemble learning is used to reduce bias and improve the robustness of the Transformer by training multiple models with the same architecture but different parameters and averaging their outputs to choose the top-N precursors. All precursors obtained will be input to the one-step models iteratively to get the further precursors until the pre-defined building blocks are obtained. While the time consumption will increase exponentially as the step grows, Retro* was conducted to select the top branches for each step (i.e. only part of precursors will be preserved) in the iteration. This process is reflected by the red cross on the unreasonable reaction in retro* part of Figure 1B (see below). We have modified the Figure 1B to make it clearer. The description in lines 108 to 120 has also been reorganized in the revised manuscript as shown below:

“Herein, we present BioNavi-NP as a new tool to propose NP biosynthetic pathways from simple building blocks in an optimal fashion. As depicted in Figure 1b, we first trained Transformer neural networks to generate the candidate precursors for a target NP. Through data augmentation and ensemble learning, our best model achieved a top-10 prediction accuracy of 60.6% on the single-step biosynthetic test set, 1.4 times more accurate than the previous rule-based and sequence-based deep learning models. Based on the single-step model, we further developed an automatic retro-biosynthesis route planning system (BioNavi-NP) through the deep learning-guided AND-OR tree-based searching algorithm, which can solve the combinatorial number of options caused by the branches of the synthetic pathway. As a result, BioNavi-NP successfully identified biosynthetic pathways for 90.2% of 368 test compounds and recovered the building blocks for 72.8%, demonstrating its potential for bio-retrosynthetic pathway elucidation or reconstruction. For each biosynthetic step in the multi-step bio-retrosynthetic routes, we further evaluated the plausible enzymes through Selenzyme, an enzyme prediction tool. All outputs of the BioNavi-NP can be visualized by an interactive website (<http://biopathnavi.qmclab.com/>), where the predicted reaction pathways are sorted by the computational cost, length, and organism-specific enzyme.”

i) Biosynthetic Planning with Retro*

ii) Ensemble

iii) Transformer

2) Lines 230-239: Given the abundance of metabolic products created from amino acids, it is surprising that amino acids have the lowest correct pathway and building block scores. How does the performance metrics in Figure 2b compare to the abundance of training data (e.g. what fraction of the training data corresponded with each category of NP). Is this truly a lack of data problem (suggested in line 239), an issue of difficult decomposition (suggested in lines 235 to 238), or attributable to both?

- We note that the category of internal test set shown in Figure 2B was assigned **manually** based on their ground-truth pathways, it is difficult to categorize all of the structures in the training set. To address the issues, we have further analyzed the datasets and observed that: i) Statistically, out of a total of 25 internal test cases with multiple building blocks, 64% (16/25) belonged to the AAs pathway, proving our statements that NPs from the AAs biosynthetic pathway are more likely consist of multiple building blocks; ii) most of the reactions (89%) in our in our training dataset are singletons (i.e. only one reactant and one product), which makes it more difficult for our model to decompose some complex NPs especially those from the AAs pathway with multiple building blocks.

In summary, the low hit rates of NPs from amino acids pathway can be attributed to both of these reasons.

3) Assessment of the tool: In some places, the assessment criteria for the tool seem to be low bars that would be still impractical to implement in vivo or requires a lot of test runs in vitro. For instance, in lines 241 to 250, in 68% of cases, at least one building block was correctly identified. This seems like a low threshold considering that three or more building blocks are needed for many of the case studies. What are the rates for all building blocks being correct? If this rate is low, why would it be advantageous to have a system like BioNavi-NP which might only identify one correct precursors across five or more different synthesis pathways, but will still need experimental validation? I think this issue needs to be addressed in the manuscript.

- The rates for all building blocks being correct is 44% and 8% (11/25 and 2/25) for BioNavi-NP and RetroPathRL, respectively. This shows that our model has advantages in

complex structures, despite the fact that intermediates can be incorrect or missing in the output of BioNavi-NP. While we want to emphasize that for most cases, the incomplete results can also be practical to experimental test. Taking case 1 and 2 (see below, also depicted in Supplementary Figure 6) as examples, the dimethylallyl diphosphate (DMAPP, highlighted in red box) is one of the building blocks in both cases, and it is missing in the outputs of BioNavi-NP. While it is not difficult to infer the missing or right part (here the DMAPP) according to the already predicted steps (step 1 in case 1, step 1 and 2 in case 2). A similar situation will occur when single amino acid or glycosyl is attached to the target structure.

We have discussed this issue in the revised Discussion section as

“Additionally, intermediates may be incorrect or missing in some predictions. Even so, the results may also be practical to experimental test for many cases, since the missing parts can be inferred by the predicted steps (such as the dimethylallyl diphosphate in case 1 and 2, Supplementary Figure 6), or they can be existed in other output pathways (e.g. the fourth and fifth candidate constitute the complete biosynthetic pathway of sterhirsutin J, Figure 3b).”

4) Discussion. While a frank discussion of the limitation of the current tool is appreciated, this discussion section appears to lack much of a review of the improvements that BioNavi-NP brings to this area of research and seems to focus too much on the limitations. Consider the phrase on lines 346 to 347: “BioNavi-NP seems still inadequate”, which seems to beg the question, if the tool is inadequate, why publish it now and in such a prestigious journal? As I have said before, I believe the work holds great potential, but the lack of credit given to the tool in the discussion section seems to undercut its inclusion in such a prestigious journal as Nature Communications. Suggest rewriting.

➤ Thank you for the suggestion. We have removed the sentence of “BioNavi-NP seems still inadequate” and re-written the whole Discussion section. Please refer to the revised manuscript.

5) Energy costs. In many organisms, the energy costs associated with NP production either limit its production (e.g. not enough energy to produce something) or drive its production (e.g. the production of the NP serves as an electron or energy dump). Does BioNavi-NP consider or predict cofactors necessary for each predicted biosynthesis step?

➤ That is a good point. Considering the intrinsic drawback of the neural network approaches caused by the SMILES input, we had made efforts to remove the majority of cofactors from

the reactions to avoid too many by-products/cofactors affecting the syntax of SMILES. Hence, BioNavi-NP only considers the transformation of the main components of the biosynthesis steps. Although we agree that the energy cost can be an important element for the biosynthesis planning in a specific organism, the mass-unbalanced nature of the data set (also mentioned in the response to reviewer 1) limits our ability to make good use of the co-factor for prediction. Thus, we leave it in the future research.

Minor Concerns:

1) Lines 218-221: This reads that for a set of desired natural products, BioNavi-NP on average produces more alternative pathways for synthesis than does RetroPathRL (up to a maximum of 5). Then it says that BioNavi-NP has a longer pathway than does RetroPathRL in its predictions. The claim is then made that BioNavi-NP is able to more effectively address complex NPs. This last claim does not appear to be strictly true for this particular comparison. This shows only that BioNavi-NP can produce more hypothetical synthesis pathways with more complexity than RetroPathRL, not that it is more efficient. A “more efficient” claim would need to be justified with computing time required to get the results. Also unclear here is how many NPs were used for this analysis.

➤ We agree, the “more efficient” claim has been removed and the sentence has been revised. Here the analysis was based on 368 internal cases.

2) Web interface. Does the web interface allow selection of precursor metabolites, or are these pre-defined?

➤ A core library (20 building blocks) and an extended library (437 available precursor metabolites extracted from iML1515 model) were pre-defined in the web server. Now we allow users to add new precursors, while this may not work well if the new precursors are not (or rarely) seen in our training set. We have added this notice in the corresponding place of web interface.

3) Tool. Does the tool consider non-enzymatic steps?

➤ Yes, the non-enzymatic reactions from USPTO were integrated to the training set as described in the manuscript.

Concerns of grammar, clarity, spelling, and similar:

1) Citations. The in-text citation style is somewhat inconsistent, in that in line 65 the citation occurs after the period, yet in line 66 the citation occurs before the period. Please make consistent.

➤ Revised accordingly.

2) Proofreading. There are still a few areas in need of proofreading, for example (line 69): “commercially-relevant quantities of NPs is a major obstacle to their therapeutic translation.” Should be “commercially-relevant quantities of NPs are a major obstacle to their therapeutic translation.” Another example is line 92: “predicted reaction cannot happen even it obeys the reaction rule”, should be “predicted reaction cannot happen even if obeys the reaction rule”. Spelling mistakes also are still present, such as in line 119 “network”. Formatting mistakes are also present such as in line 338 where there is a period on either side of a reference. Please make sure

to proofread your article again.

- Thank you very much for the comment. We have corrected all that you suggested and done a new proofreading. The updated manuscript has further been polished by a native English-speaking expert in cheminformatics.

Reviewers' Comments:

Reviewer #1:

Remarks to the Author:

The language has substantially improved, and most points raised by the reviewers previously have been addressed in the new version of the manuscript. In general, the proposed deep learning based approach looks promising. However, the results do not demonstrate substantial improvement over existing methods. The comparison with existing methods seems biased, and as the rules-based methods are excluded from comparison as a class. I think for reader it would be still interesting to know what would be the benefit of using the BioNavi-NP in comparison to other existing methods for pathway search, independently on whether ML-based or rules-based.

Major:

BioNavi-NP has increase in top-10 accuracy in comparison to RetroPathRL of 18.5% (42.1% to 60.6%) and only 1% improvement in term of top-1 accuracy (page 6). Authors demonstrate that these results demonstrate the power of deep learning, however, it looks like RetroPathRL and reinforcement learning would be actually an approach of choice for many groups interested to find pathway design solutions, as it has comparable accuracy and just 2h execution time in comparison with 18h for BioNavi-NP (Table 2). The number of hours stated in the Table 2 does not match the following sentence on page 10:

Note that RetroPathRL requires more than 10000 iterations and takes 9 times longer than BioNavi-NP (123 hours for RetroPathRL vs. 13.5 hours for BioNavi-NP).

Please, try to optimize your algorithm to achieve better execution time. Alternatively, you could define a niche in which your method would indeed have significant advantages in comparison to other existing methods.

The biggest success of this method is 90.2% success rate for pathways identification. However, as authors state in the discussion, major issues remain with the predicted pathways due to the scoring function, which does not include yield, enzyme availability, flux balance and/or any other metrics that in fact are standard for the community. In the discussion, authors state that "anomalous data leading to unjustified shortcuts", "shortcuts through small cofactor-type molecules or by-products" (page.13) remain an issue for some fraction of their results. Therefore, it is hard to judge whether the main result of the article contains the sensible pathways and which fraction of 90.2% is "anomalous data". Methods to overcome this issue based on atom-mapping exist in the community (which authors mention as a subject for future work).

The Selenzyme RESTful service that authors have integrated remains "503 Service Temporarily Unavailable", therefore, unless Selenzyme authors fix the service, the feature of enzyme prediction that is crucial for the predicted pathways implementation will remain "Failed to submit! Selenzyme service is temporarily unavailable." Please, provide a backup.

In general, I could only see the significant improvement and the value for the community in the building block identification, where the significant improvement was indeed achieved.

Minor:

Some minor issues remain to improve the clarity of the methods and results.

Page 8:

"It should be noted that RetroPathRL contains the reaction rules extracted from MetaNetX, while all the internal test NPs were included in MetaNetX."

-> Since the reaction rules are extracted from reactions, and NPs are compounds, I do not think that this note demonstrates any bias towards MetaNetX coverage.

Page 8:

"When selecting the output option as top-5 (see Table 2), BioNavi-NP predicted an average of 4.9 pathways, higher than the 2.8 by RetroPathRL. "

-> This sentence seems unnecessary to me - There is no point in comparing the number of solutions you get from an algorithm that takes the number of solutions as an input with an algorithm where the number of solutions is part of the output.

Page 8:

"Meanwhile, BioNavi-NP achieved 56.0% and 24.7% for the hit rates of building blocks and pathways, remarkably outperforming RetroPathRL (4.8% and 3.8%), respectively."

-> Again, RetroPathRL was not designed for building blocks prediction, therefore, such comparison does not bring much value, you can state it as an independent feature that was not considered in RetroPathRL.

Page 10:

"but the default settings and building blocklist can be modified as needed"

Blocklist is the list of items to exclude. Did you mean: list of building blocks?

Page 15:

"Different from organic synthesis and metabolic engineering, we only assigned 40 building blocks (by default, extended and user-defined library are also available) from three major sources amino acids, organic acids and other molecules upstream the biosynthetic pathway of natural products (Supplementary Figure 3), which have been proved to be the basic components of most of the natural products."

Please, provide citation

Supplementary material:

p.7, supplementary table 1, BNICE.ch webservice latest update (ATLASx) data source is broader than just KEGG.

p.11. "A interactive web page": should be "An interactive"

p.13, 14, 15 etc. "number of pathway was set to": should be "number of pathways"

Reviewer #2:

Remarks to the Author:

The authors have addressed most of my remaining comments.

Few minor issues to still sort out:

1. Abstract: "only fewer than 30,000 [natural products] have been reported to be involved in a total of about 33,000 enzyme-catalyzed reactions." This statement is ambiguous. It does not convey the same information as provided (correctly) in the Introduction section: "only about 33,000 enzymatic reactions have been characterized and confirmed, corresponding to fewer than 30,000 NPs serving as a substrate or product".
2. Abstract: "1.7 times more accurate than the conventional rule-based approaches". Use of "the" indicates to the reader that the authors have tested every single, rule-based approach in existence, which is not the case. Hence "the" should be replaced by something like "existing".
3. The Selenzyme web service has been offline for many days now. The service may have been discontinued. Can the Authors please investigate and adjust their manuscript and web service

accordingly?

Reviewer #3:

Remarks to the Author:

Title: BioNavi-NP: Biosynthesis Navigator for Natural Products

Authors: Shuangjia Zheng, Tao Zeng, Chengtao Li, Binghong Chen, Connor W. Coley, Yuedong Yang, Ruibo Wu

Recommendation: Accept with minor revisions

General reviewer comments: As before, I think this tool shows great promise for the development of synthetic pipelines for NP synthesis and it holds great potential through the identification of system pathways. Since the last revision, the TOC graphic is much improved, the writing is more clear, and the revised image (1B) is easier to understand. Overall, in my opinion, this manuscript is close to publication.

Major Concerns:

1) none

Minor Concerns:

1) Recovered pathways and building blocks (comment made throughout the manuscript). When it is says that "BioNavi-NP could identify biosynthetic pathways for 90.2% of 368 tests and recover the building blocks for 72.8%", this could easily lead to a misunderstanding. For instance, I understand that this means that 90.2% of compounds had some pathway prediction and 72.8% had at least one correct building block recovered. Is this correct? Another interpretation could easily be that 90.2% of test compounds have a predicted pathway but only 72.8% had predicted building blocks. Which leaves a 17.4% gap of metabolites that somehow have pathways but not building blocks input to those pathway. But how would a compound have a synthesis pathway and not building blocks for those pathways. The way this is phrased is a little weird or vague and could be worded better to avoid confusion.

2) Lines 253: When something is claimed as "essentially the same" a statistical test would be in order. Since it is a yes-no result, a chi squared test would suffice. If no statistical difference exists, then this statement can be made.

3) Discussion. The rebuttal mentioned work related to cofactors in future, but not present in the discussion. I think this weakness of the current method (e.g. lack of cofactor consideration) needs to be acknowledge in the discussion, followed up with an acknowledgement that this is part of the future research direction for this work.

Concerns of grammar, clarity, spelling, and similar:

1) Line 345. The phrase "outside of knowledge" doesn't sound right. Perhaps something more like "outside of current knowledge", or "currently unknown reactions" or similar.

Reviewer #1 (Remarks to the Author):

The language has substantially improved, and most points raised by the reviewers previously have been addressed in the new version of the manuscript. In general, the proposed deep learning based approach looks promising.

➤ Thanks for the reviewer's positive comments.

However, the results do not demonstrate substantial improvement over existing methods. The comparison with existing methods seems biased, and as the rules-based methods are excluded from comparison as a class. I think for reader it would be still interesting to know what would be the benefit of using the BioNavi-NP in comparison to other existing methods for pathway search, independently on whether ML-based or rules-based.

➤ We'd like to clarify that both RetroPathRL and RetroPath2.0 are rule-based method for pathway search. Thus, in the manuscript we did compare BioNavi-NP head-to-head with other methods including ML-based (seq2seq) and rule-based (RetroPathRL, RetroPath2.0) methods. The results showed that we consistently outperformed these methods in multiple metrics (Table 1, Table 2 and SI Table 3). We didn't include other rule-based methods because they do not make code/software available (e.g., PathPred, Envipath) or do not support automatic **large-scale** benchmarking tests (e.g., Novostoc, RouteSearch).

Major:

BioNavi-NP has increase in top-10 accuracy in comparison to RetroPathRL of 18.5% (42.1% to 60.6%) and only 1% improvement in term of top-1 accuracy (page 6). Authors demonstrate that these results demonstrate the power of deep learning, however, it looks like RetroPathRL and reinforcement learning would be actually an approach of choice for many groups interested to find pathway design solutions, as it has comparable accuracy and just 2h execution time in comparison with 18h for BioNavi-NP (Table 2). The number of hours stated in the Table 2 does not match the following sentence on page 10:

Note that RetroPathRL requires more than 10000 iterations and takes 9 times longer than BioNavi-NP (123 hours for RetroPathRL vs. 13.5 hours for BioNavi-NP).

Please, try to optimize your algorithm to achieve better execution time. Alternatively, you could define a niche in which your method would indeed have significant advantages in comparison to other existing methods.

➤ Sorry for the confusion. We'd like to emphasize that the top-10 accuracy is essential for the final retro-biosynthesis planning because the planning algorithm will combine the single-step models to form the best route. The top-1 accuracy isn't important because the predicted top single-step reactions aren't always selected, which is why we have set default expansion parameters to 10 instead of 1 (Page 7). Furthermore, the inconsistency in running times is due to the different settings of iteration parameter. As shown in Table 2 for the final results, when using RetroPathRL's default iteration setting (1000 iterations), it required only 2 hours but achieved a low building block hit rate of 4.8% and a pathway hit rate of 3.8% on the internal test set, compared to 56.0% and 24.7% by BioNavi-NP. To make a fair comparison with similar running times, we employed the reported settings of RetroPathRL and performed 10000~15000

iterations for each compounds on their dataset with 152 compounds. As shown in Table S3, RetroPathRL costs 123 hours, ~9 times longer than BioNavi-NP (13.5 hours), but their accuracy (83.6%) is still lower than our method (94.7%). We have also optimized the algorithm on our server, and the average running time decreases from ~5 to 3.5 minutes for a single compound that is acceptable for biochemists. By comparison, RetroPathRL's prediction with 10000 iterations requires ~28 minutes. We have clarified these in the text:

Page 7 “*all deep learning models were limited to 100 iterations and 10 expansions, while RetroPathRL was set to 1000 iterations and 10 expansions following its default settings. Note that the number of expansion represents the top-N metabolites that the model will predict in every single step, i.e., top-N in Table 1.*”

Page 10 “*Note that to achieve these results, RetroPathRL needed to increase the iteration numbers from 1000 to 10000~15000, and thus takes 9 times longer than BioNavi-NP (123 hours for RetroPathRL vs. 13.5 hours for BioNavi-NP).*”

The biggest success of this method is 90.2% success rate for pathways identification. However, as authors state in the discussion, major issues remain with the predicted pathways due to the scoring function, which does not include yield, enzyme availability, flux balance and/or any other metrics that in fact are standard for the community. In the discussion, authors state that “anomalous data leading to unjustified shortcuts”, “shortcuts through small cofactor-type molecules or by-products” (page.13) remain an issue for some fraction of their results. Therefore, it is hard to judge whether the main result of the article contains the sensible pathways and which fraction of 90.2% is “anomalous data”. Methods to overcome this issue based on atom-mapping exist in the community (which authors mention as a subject for future work).

➤ We thank the reviewer's suggestions. We have addressed these issues in alternative ways: to make up the intrinsic shortage of scoring function of our model, we have added the Selenzyme and the annotation modules to predict species-based enzyme availability and offer reactions/compounds information for pathway re-ranking. To handle the cofactor-type molecules and by-products, the maximum common substructure match algorithm has been implemented to reduce the ambiguous role of generic compounds as described in the Method section. We'd like to emphasize that though atom-mapping strategies are often used in rule-based methods, such methods haven't been applied to the **rule-free** approach in biosynthesis, likely due to the model incompatibility and the complexity of biochemical reaction.

We have clarified this in the revised manuscript as: “*Future work on integrating atom-mapping approaches into the rule-free models will allow better mass-balance for the single-step retro-biosynthesis predictions.*”

The Selenzyme RESTful service that authors have integrated remains “503 Service Temporarily Unavailable”, therefore, unless Selenzyme authors fix the service, the feature of enzyme prediction that is crucial for the predicted pathways implementation will remain “Failed to submit! Selenzyme service is temporarily unavailable.” Please, provide a backup.

➤ Thank you for the kind advice, we have contacted the administrator of Selenzyme server. Their server was under maintenance for a period of time and now it is running again (see below). Additionally, to facilitate the enzyme prediction, we have added an alternative enzyme

prediction tool E-zyme 2 (see below). Users can freely use both of them in the web interface. This has been also stated in the revised manuscript as: “We introduced Selenzyme to search for candidate enzymes for specific reactions because of its lightweight RESTful service and additional biological metadata. As a result, pathways output by BioNavi-NP can be further ranked by reaction similarity and the taxonomic distance between the organism of enzyme and user-defined organism. In addition, the hyperlink and required input chemical information of E-zyme 2 are also provided for enzyme prediction as an alternative.”

Dear Zeng,

It is nice to know the tool was useful. I will check as we are maintaining this tool but also looking at the hosting of it.

Best wishes
Ros

Dr Rosalind Le Feuvre
Director of Operations, SYNBIOCHEM and Future BRH
Manchester Institute of Biotechnology,
University of Manchester
M13 9PL

0161 3065184

Dear Zeng Tao,

Thank you for your interest in Selenzyme and for contacting us about the problem. I am happy to let you know that the tool is now running.
<http://selenzyme.synbiochem.co.uk/>

Please let me know if you have any further issues.

Search enzyme for reaction (Provided by Selenzyme and E-zyme 2)

2. Click to copy the MOL file of reactant and product, respectively (used for E-zyme 2)

Selenzyme E-zyme 2 1. Click to open a new tab redirecting to E-zyme 2

Users need to submit the job by the Web interface of E-zyme 2 with copying the MOL files above

In general, I could only see the significant improvement and the value for the community in the building block identification, where the significant improvement was indeed achieved.

- We thank you for the endorsement in the task of building block identification. We would like to emphasize that, as a precedent work that uses rule-free methods for predicting NPs' biosynthesis pathways, it definitely has both pros and cons. The partial loss in pathway verifiability/interpretability brings the possibility of generalization to new complex natural product. In this sense, as reviewer 3 states, this work “holds great potential through the identification of system pathways.” In addition, we have made the data, source code and server publicly available, which will accelerate the machine learning community and the biosynthesis community to refine and iterate on the tool. We hope this response shows you more of the benefits of this work.

Minor:

Some minor issues remain to improve the clarity of the methods and results.

Page 8:

“It should be noted that RetroPathRL contains the reaction rules extracted from MetaNetX, while all the internal test NPs were included in MetaNetX.”

-> Since the reaction rules are extracted from reactions, and NPs are compounds, I do not think that this note demonstrates any bias towards MetaNetX coverage.

- We mentioned this because the full coverage of MetaNetX makes the comparison favorable for RetroPathRL as the reaction information of the test NPs has been already included in their rule library (known as data leakage in machine learning).

Page 8:

“When selecting the output option as top-5 (see Table 2), BioNavi-NP predicted an average of 4.9 pathways, higher than the 2.8 by RetroPathRL. “

-> This sentence seems unnecessary to me - There is no point in comparing the number of solutions you get from an algorithm that takes the number of solutions as an input with an algorithm where the number of solutions is part of the output.

- This has been revised to “*When selecting the output option as top-5 (see Table 2), BioNavi-NP predicted an average of 4.9 pathways*”

Page 8:

“Meanwhile, BioNavi-NP achieved 56.0% and 24.7% for the hit rates of building blocks and pathways, remarkably outperforming RetroPathRL (4.8% and 3.8%), respectively.”

-> Again, RetroPathRL was not designed for building blocks prediction, therefore, such comparison does not bring much value, you can state it as an independent feature that was not considered in RetroPathRL.

- Thank you for the suggestion. Our approach, similar to RetroPathRL, is not specifically designed for building blocks prediction. Note that both these two models do not obtain supervision or reward directly from the building blocks during training and testing. Instead, the building blocks are only used as the end points for the pathway search. The increases in the building block hit rates are due to a combination of improvements in single-step prediction and search algorithm. Moreover, the hit rate of building block is a key indication of whether the pathway search is going in the right direction, and reflects the quality of the predicted pathways to some degree. Therefore, we believe that this comparison is meaningful.

Page 10:

“but the default settings and building blocklist can be modified as needed”

Blocklist is the list of items to exclude. Did you mean: list of building blocks?

- Revised accordingly.

Page 15:

“Different from organic synthesis and metabolic engineering, we only assigned 40 building blocks

(by default, extended and user-defined library are also available) from three major sources amino acids, organic acids and other molecules upstream the biosynthetic pathway of natural products (Supplementary Figure 3), which have been proved to be the basic components of most of the natural products.”

Please, provide citation

➤ Added accordingly.

Supplementary material:

p.7, supplementary table 1, BNICE.ch webservice latest update (ATLASx) data source is broader than just KEGG.

p.11. “A interactive web page”: should be “An interactive”

p.13, 14, 15 etc. “number of pathway was set to”: should be “number of pathways”

➤ Thank you for the comments. The issues above have been addressed accordingly.

Reviewer #2 (Remarks to the Author):

The authors have addressed most of my remaining comments.

Few minor issues to still sort out:

1. Abstract: “only fewer than 30,000 [natural products] have been reported to be involved in a total of about 33,000 enzyme-catalyzed reactions.” This statement is ambiguous. It does not convey the same information as provided (correctly) in the Introduction section: “only about 33,000 enzymatic reactions have been characterized and confirmed, corresponding to fewer than 30,000 NPs serving as a substrate or product”.

➤ Thank you for pointing out this, the sentence in the Abstract section has been revised to “*only about 33,000 enzymatic reactions have been characterized and confirmed, corresponding to fewer than 30,000 NPs serving as a substrate or product*”.

2. Abstract: “1.7 times more accurate than the conventional rule-based approaches”. Use of “the” indicates to the reader that the authors have tested every single, rule-based approach in existence, which is not the case. Hence “the” should be replaced by something like “existing”.

➤ Revised accordingly.

3. The Selenzyme web service has been offline for many days now. The service may have been discontinued. Can the Authors please investigate and adjust their manuscript and web service accordingly?

➤ Thank you for the kind advice, we have contacted the administrator of Selenzyme server. Their server was under maintenance for a period of time and now it is running again (see below). Additionally, to facilitate the enzyme prediction, we have added an alternative enzyme prediction tool E-zyme 2 (see below). Users can freely use both of them in the web interface. This has been also stated in the revised manuscript as: “*We introduced Selenzyme to search for*”

candidate enzymes for specific reactions because of its lightweight RESTful service and additional biological metadata. As a result, pathways output by BioNavi-NP can be further ranked by reaction similarity and the taxonomic distance between the organism of enzyme and user-defined organism. In addition, the hyperlink and required input chemical information of E-zyme 2 are also provided for enzyme prediction as an alternative.”

Dear Zeng,

It is nice to know the tool was useful. I will check as we are maintaining this tool but also looking at the hosting of it.

Best wishes
Ros

Dr Rosalind Le Feuvre
Director of Operations, SYNBIOCHEM and Future BRH
Manchester Institute of Biotechnology,
University of Manchester
M13 9PL

0161 3065184

Dear Zeng Tao,

Thank you for your interest in Selenzyme and for contacting us about the problem. I am happy to let you know that the tool is now running.
<http://selenzyme.synbiochem.co.uk/>

Please let me know if you have any further issues.

Search enzyme for reaction (Provided by Selenzyme and E-zyme 2)

2. Click to copy the MOL file of reactant and product, respectively (used for E-zyme 2)

1. Click to open a new tab redirecting to E-zyme 2

Users need to submit the job by the Web interface of E-zyme 2 with copying the MOL files above

Reviewer #3 (Remarks to the Author):

General reviewer comments: As before, I think this tool shows great promise for the development of synthetic pipelines for NP synthesis and it holds great potential through the identification of system pathways. Since the last revision, the TOC graphic is much improved, the writing is more clear, and the revised image (1B) is easier to understand. Overall, in my opinion, this manuscript is close to publication.

➤ We appreciate the review's positive comments.

Minor Concerns:

1) Recovered pathways and building blocks (comment made throughout the manuscript). When it says that “BioNavi-NP could identify biosynthetic pathways for 90.2% of 368 tests and recover the building blocks for 72.8%”, this could easily lead to a misunderstanding. For instance, I understand that this means that 90.2% of compounds had some pathway prediction and 72.8% had at least one correct building block recovered. Is this correct? Another interpretation could easily be that 90.2% of test compounds have a predicted pathway but only 72.8% had predicted building

blocks. Which leaves a 17.4% gap of metabolites that somehow have pathways but not building blocks input to those pathway. But how would a compound have a synthesis pathway and not building blocks for those pathways. The way this is phrased is a little weird or vague and could be worded better to avoid confusion.

- Thanks for the suggestions, your understanding is right and we have changed the description to *“BioNavi-NP successfully identified biosynthetic pathways for 90.2% of 368 test compounds and recovered the reported building blocks as in the dataset for 72.8%”*.

2) Lines 253: When something is claimed as “essentially the same” a statistical test would be in order. Since it is a yes-no result, a chi squared test would suffice. If no statistical difference exists, then this statement can be made.

- “essentially” has been changed to “roughly”.

3) Discussion. The rebuttal mentioned work related to cofactors in future, but not present in the discussion. I think this weakness of the current method (e.g. lack of cofactor consideration) needs to be acknowledge in the discussion, followed up with an acknowledgement that this is part of the future research direction for this work.

- We have acknowledged the related content in the discussion as *“Furthermore, a few predicted pathways tend to take shortcuts through small cofactor-type molecules or by-products. In fact, efforts have been made in data pre-processing to reduce the ambiguous role of generic compounds, but there are still a small fraction of anomalous data leading to unjustified shortcuts. Future work to combine atom-mapping approaches will allow better mass-balance for the single-step prediction.”*

Concerns of grammar, clarity, spelling, and similar:

1) Line 345. The phrase “outside of knowledge” doesn’t sound right. Perhaps something more like “outside of current knowledge”, or “currently unknown reactions” or similar.

- Revised accordingly.

Reviewers' Comments:

Reviewer #1:

Remarks to the Author:

The proposed ML-based "rule-free" method has certain novelty that probably will be interesting to the readers interested in the interface between ML and pathway design. The authors have addressed our comments. Some of our concerns they decided to leave for the future research. They decided not to do the major revision that we suggested and provided some justifications that seem to be acceptable. We still believe the method would benefit from the improvements suggested in the previous round of review, and we hope the authors will address it in their future publications. For the future research, we would also suggest that authors validate and compare their method not only with RetroPath, but with some other state of the art methods in the field, such as enviPath, ATLASx, etc. The overall suggestion would be to accept the article.

Reviewer #2:

None

Reviewer #3:

Remarks to the Author:

All of my questions or concerns on this article have been addressed by the authors through these several rounds of review and I have no further comments/concerns.

- We thank the reviewers for their constructive suggestions in the previous round of review. In the following, we provide response to the comments of Reviewer #1.

Reviewer #1 (Remarks to the Author):

The proposed ML-based “rule-free” method has certain novelty that probably will be interesting to the readers interested in the interface between ML and pathway design. The authors have addressed our comments. Some of our concerns they decided to leave for the future research. They decided not to do the major revision that we suggested and provided some justifications that seem to be acceptable. We still believe the method would benefit from the improvements suggested in the previous round of review, and we hope the authors will address it in their future publications. For the future research, we would also suggest that authors validate and compare their method not only with RetroPath, but with some other state of the art methods in the field, such as enviPath, ATLASx, etc. The overall suggestion would be to accept the article.

- We thank the reviewer for the positive feedback and constructive suggestions. We agree with the reviewer that the approaches like atom-mapping can be used to improve the data quality and we will explore their feasibility in the deep learning model with biochemical reactions in the future work. Other tools which are available will also be included for comparison.